# A large accessory protein interactome is rewired across environments

**Zhimin Liu[1,2], Darach Miller[3,4], Fangfei Li[2,5], Xianan Liu[1,2], Sasha F Levy[1,2,3,4,5,6]\***

[1]Department of Biochemistry, Stony Brook University, Stony Brook, United States; [2]Laufer Center for Physical and Quantitative Biology, Stony Brook University, Stony Brook, United States; [3]Joint Initiative for Metrology in Biology, Stanford, United States; [4]Department of Genetics, Stanford University, Stanford, United States; [5]Department of Applied Mathematics and Statistics, Stony Brook University, Stony Brook, United States; [6]SLAC National Accelerator Laboratory, Menlo Park, United States

**Abstract** To characterize how protein-protein interaction (PPI) networks change, we quantified the relative PPI abundance of 1.6 million protein pairs in the yeast *Saccharomyces cerevisiae* across nine growth conditions, with replication, for a total of 44 million measurements. Our multi-condition screen identified 13,764 pairwise PPIs, a threefold increase over PPIs identified in one condition. A few 'immutable' PPIs are present across all conditions, while most 'mutable' PPIs are rarely observed. Immutable PPIs aggregate into highly connected 'core' network modules, with most network remodeling occurring within a loosely connected 'accessory' module. Mutable PPIs are less likely to co-express, co-localize, and be explained by simple mass action kinetics, and more likely to contain proteins with intrinsically disordered regions, implying that environment-dependent association and binding is critical to cellular adaptation. Our results show that protein interactomes are larger than previously thought and contain highly dynamic regions that reorganize to drive or respond to cellular changes.

## Introduction

As environmental conditions change, cells undergo programmed alterations that ultimately rewire PPI networks to execute different biological processes. Numerous examples of localized PPI rewiring have been found (*Balajee and Geard, 2001*; *Celaj et al., 2017*; *Mailand et al., 2013*; *Marles et al., 2004*; *Rochette et al., 2014*). Yet, little is known about how PPI networks reorganize on a global scale or what drives these changes. One challenge is that commonly-used high-throughput PPI screening technologies are geared toward PPI identification (*Gavin et al., 2002*; *Ito et al., 2001*; *Tarassov et al., 2008*; *Uetz et al., 2000*; *Yachie et al., 2016*; *Yu et al., 2008*), not a quantitative analysis of relative PPI abundance that is necessary to determine if changes in the PPI network are occurring. The murine dihydrofolate reductase (mDHFR)-based protein-fragment complementation assay (PCA) provides a viable path to characterize PPI abundance changes because it is a sensitive test for PPIs in the native cellular context and at native protein expression levels (*Freschi et al., 2013*; *Remy and Michnick, 1999*; *Tarassov et al., 2008*). Indeed, moderate-scale mDHFR-PCA studies have characterized the dynamics of a subset of known PPIs in yeast, finding that between 15% and 55% change across environments and are frequently driven by changes in protein abundance (*Celaj et al., 2017*; *Rochette et al., 2014*; *Schlecht et al., 2012*; *Schlecht et al., 2017*).

However, because mDHFR-PCA studies have only considered the dynamics of PPIs that have been identified under standard laboratory growth conditions (rich complete media), they may be providing an incomplete view of global PPI rewiring (*Celaj et al., 2017*). One possibility is that PPIs identified from a single condition are biased toward 'immutable' PPIs that are found in all conditions,

\*For correspondence:
sflevy@stanford.edu

and 'mutable' PPIs that are present in only some conditions have a large impact on global PPI network dynamics. If true, mutable PPIs would also be expected to be underrepresented in current PPI networks, with important consequences to our understanding of how the protein interactome is organized and how proteins typically interact. Previous work supports the idea that proteins that participate in mutable PPIs have different properties. For example, protein hubs predicted by mRNA co-expression data to participate in mutable PPIs ('date' hubs) have been found to have more genetic interactions and to bridge tightly connected modules in the PPI network (*Han et al., 2004*). However, the robustness of conclusions drawn from co-expression data has undergone a vigorous debate (*Agarwal et al., 2010*; *Batada et al., 2006*; *Batada et al., 2007*; *Bertin et al., 2007*; *Yu et al., 2008*).

Here, we combine the mDHFR-PCA assay with a double barcoding system (*Liu et al., 2019*; *Schlecht et al., 2017*) to quantify the relative *in vivo* PPI abundance of 1.6 million protein pairs across nine growth conditions. We find that a large majority of PPIs detected in our screen have not been identified as PPIs in standard growth conditions, providing us with a new view of how mutable PPIs contribute to PPI network rewiring.

## Results

### Defining a multi-condition PPI network

In mDHFR-PCA, two proteins of interest are fused to complementary mDHFR fragments. An interaction between the proteins reconstitutes mDHFR, providing resistance to the drug methotrexate and a growth advantage that is monotonically related to the PPI abundance (*Celaj et al., 2017*; *Freschi et al., 2013*; *Rochette et al., 2014*; *Schlecht et al., 2012*; *Schlecht et al., 2017*). We have previously shown that mDHFR-PCA can be adapted into a pooled barcode sequencing assay by fusing two genomic barcodes *in vivo* using a method called PPiSeq (*Schlecht et al., 2017*). To generate a large PPiSeq library, all strains from the protein interactome (mDHFR-PCA) collection that were found to contain a protein likely to participate in at least one PPI (1742 X 1130 protein pairs) (*Tarassov et al., 2008*) were barcoded in duplicate using the double barcoder iSeq collection (*Liu et al., 2019*), and mated together in a single pool (*Figure 1A*). Double barcode sequencing revealed that the PPiSeq library contained 1.79 million protein pairs and 6.05 million double barcodes (92.3% and 78.1% of theoretical, respectively, 1741 × 1113 protein pairs), with each protein pair represented by an average of 3.4 unique double barcodes (*Figure 1—figure supplement 1*). The library was grown under mild methotrexate selection in nine environments for 12–18 generations in serial batch culture, diluting 1:8 every ~3 generations, with a bottleneck population size greater than $2 \times 10^9$ cells (Appendix 2 Table S1). Double barcodes were enumerated over 4–5 timepoints by sequencing, and the resulting frequency trajectories (Appendix 2 Table S2) were used to estimate the relative fitness (Appendix 2 Table S3) of each strain, which is a rough measure of the average PPI abundance over a growth cycle (*Figure 1B*; *Levy et al., 2015*; *Li et al., 2018*; *Schlecht et al., 2017*). We recovered a minimum of two reliable replicate fitness estimates for 1.6 million protein pairs, and downstream analysis was limited to this set. We examined the reproducibility of fitness estimation within an environment by plotting the standard deviation by the mean of replicate fitness measures for each protein pair (*Figure 1C* and *Figure 1—figure supplement 2*). We found that fitness estimates are precise for high-fitness strains (putative PPIs), but less precise for low-fitness strains, which are more subject to noise stemming from growth bottlenecks, PCR, and sequencing (*Li et al., 2018*).

To identify protein pairs that interact, we compared replicate fitness scores for each protein pair against a set of ~17,000 negative control strains that were included in the pool (*Figure 1D*, ORF x Null). Using putatively positive and negative reference sets, we empirically determined a statistical threshold for each environment with the best balance of precision and recall (positive predictive value (PPV) >61% and true positive rate >41% in SD media, *Appendix 1—table 1*). In general, protein pairs required replicated high fitness measures (mean >0.18 in SD) and low variance (standard deviation <0.02 in SD) to be identified as a PPI (*Figure 1C and D*). However, due in part to differences in the strength of methotrexate selection across environments, the minimal fitness of a PPI varied by environment (*Figure 1—figure supplement 5* and Appendix 2 Table S4). Quantitative fitness measures between different barcodes marking the same PPI within a growth pool correlate well

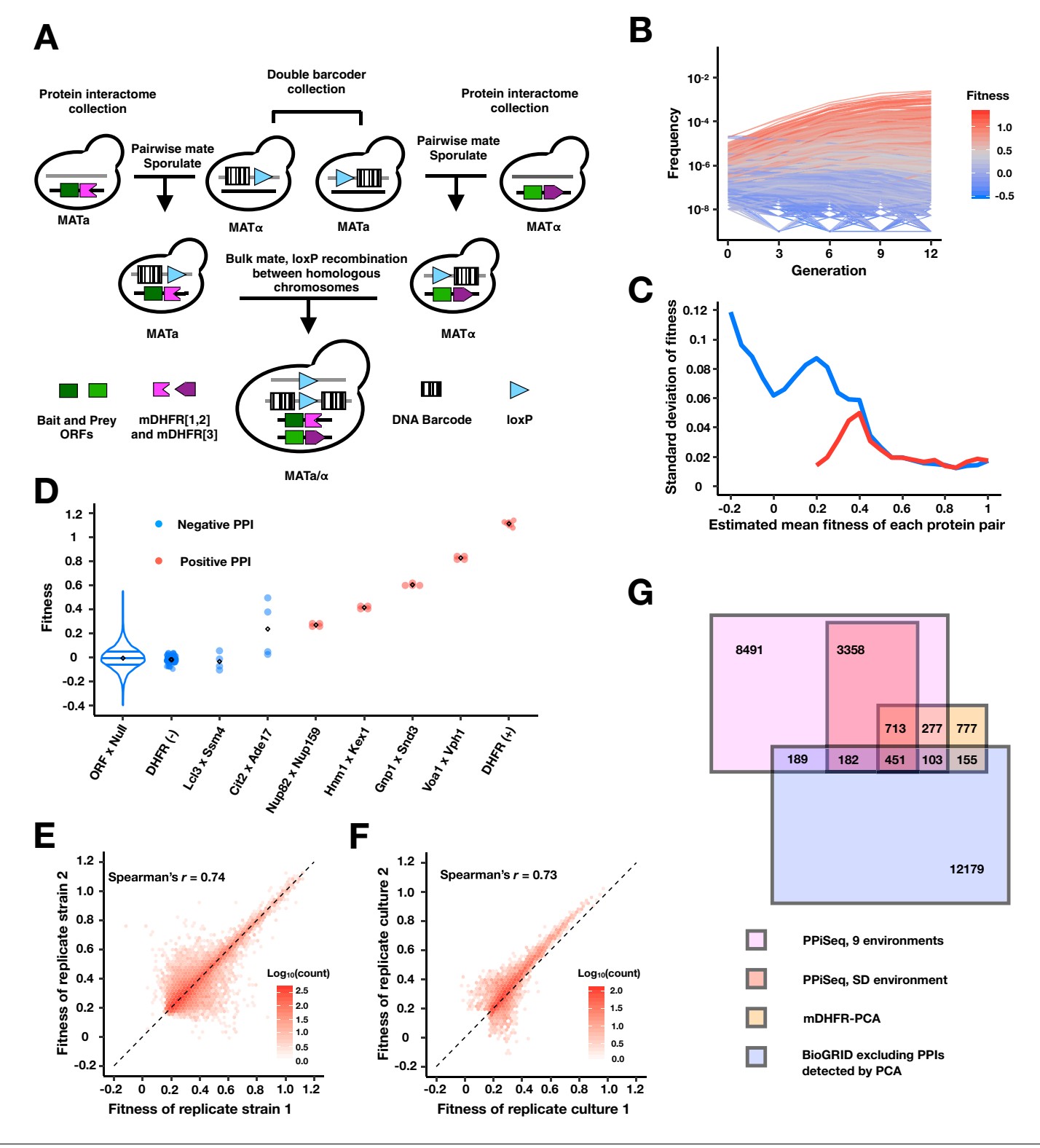

**Figure 1.** PPiSeq. (**A**) A cartoon of PPiSeq yeast library construction. Strains from the protein interactome collection are individually mated to strains from the double barcoder collection and sporulated to recover haploids that contain a mDHFR-tagged protein and a barcode. Haploids are mated as pools. In diploids, expression of Cre recombinase causes recombination between homologous chromosomes at the loxP locus, resulting in a contiguous double barcode that marks the mDHFR-tagged protein pair. (**B**) Representative double barcode frequency trajectories over twelve generations of competitive growth. Trajectories are used to calculate a quantitative fitness for each double barcoded strain. (**C**) Standard error of fitness

*Figure 1 continued on next page*

*Figure 1 continued*

estimates of protein pairs. The blue and red lines represent the median standard error for a sliding window (width = 0.05) of all fitness-ranked protein pairs and of only the positive protein-protein interactions, respectively. (D) Estimated fitness of strains with different double barcodes representing the same protein pair in the same pooled growth. Positive protein pairs are randomly selected within a fitness window. ORF x Null is a violin plot of the fitness distribution of all interactions with a mDHFR fragment that is not tethered to a yeast protein. DHFR(-) is yeast strains that lack any mDHFR fragment. DHFR(+) is yeast strains that contain a full length mDHFR under a strong promoter. (E) Density plot of the fitness of double barcodes that represent the same putative PPI in the same pooled growth. In (B–E), the data in SD environment are used. (F) Density plot of the normalized mean fitness of the same PPI between two pooled growth cultures in SD environment. PPIs detected in either one growth culture are included. (G) Venn diagram of the number of PPIs identified within our search space by PPiSeq in nine environments (magenta), PPiSeq in SD environment (pink), the interactome-scale protein-fragment complementation screen (PCA, yellow), and the BioGRID database excluding any PPIs previously detected by PCA (blue).

The online version of this article includes the following figure supplement(s) for figure 1:

**Figure supplement 1.** Double barcodes and protein pairs in the PPiSeq library.
**Figure supplement 2.** Standard error of fitness estimates of protein pairs in each environment.
**Figure supplement 3.** Density plot of the fitness of double barcodes that represent the same positive PPI in the same pooled growth of each environment.
**Figure supplement 4.** Comparison of PPiSeq data in SD condition to other PPI datasets.
**Figure supplement 5.** The OD 600 trajectories of DHFR(-) strain in various conditions with and without 0.5 µg/mL methotrexate.

(0.67 < *r* < 0.92 for all environments, Spearman's correlation, *Figure 1E* and *Figure 1—figure supplement 3*), as do fitness measures of the same PPI assayed in replicate growth cultures (Spearman's *r* = 0.73, *Figure 1F*).

In total, we identified 13,764 PPIs across nine environments, a 2.9-fold increase over PPIs identified under standard growth conditions (SD media), and a 5.6-fold increase over colony-based mDHFR-PCA (*Figure 1G* and *Figure 1—figure supplement 4*). Within our search space, PPIs identified by liquid-growth-based PPiSeq encompass 62% (1544 of 2476 PPIs, PPV > 98.2% in *Tarassov et al., 2008*, see Appendix 1 for differences between PPVs) of those identified by mDHFR-PCA, but only 7% (925 of 13,259 PPIs) of those identified by other methods. In addition, PPiSeq identified 34% of PPIs (1838 of 5347) that mDHFR-PCA had identified as likely to be PPIs (80% < PPV < 98.2% in *Tarassov et al., 2008*) but had not called.

Further highlighting similarities to colony-based mDHFR-PCA, PPiSeq is enriched for PPIs within and between membranous compartments (*Figure 2A*, blue box) and those involved in cell division (purple box), and between related biological processes (*Figure 2B*, green box). By examining how much PPIs change across environments, we found that some cellular compartments and biological processes were more likely to gain or lose PPIs. For example, the number of PPIs associated with the chromosome are more variable (*Figure 2A*, orange triangles), as are those involved in transcription (*Figure 2B*, black triangles), which is consistent with a role for PPI-level regulation of gene expression. Also changing are PPIs involved in protein translation, RNA processing, and ribosome regulation (*Figure 2B*, brown triangles), which could reflect a global regulation of ribosome production rates in different growth or stress conditions (*Brauer et al., 2008*; *Gasch et al., 2000*). However, some processes are less likely to change in PPI number, such as endocytosis, exocytosis, vacuolar organization, and transport of amino acids, lipids, carbohydrates and endosomes.

## A large dynamic accessory protein interactome

We partitioned PPIs by the number of environments in which they were identified and defined PPIs at opposite ends of this spectrum as 'mutable' PPIs (identified in only 1–3 environments) and 'immutable' PPIs (identified in 8–9 environments). Mutable PPIs far outnumbered immutable PPIs, with PPIs identified in only one environment outnumbering all other PPIs combined (*Figure 3A* and *Figure 3—figure supplement 1*). Immutable PPIs were likely to have been previously reported by colony-based mDHFR-PCA or other methods, while the PPIs found in the fewest environments were not. One possible explanation for this observation is that previous PPI assays, which largely tested in standard laboratory growth conditions, and variations thereof, are biased toward identification of the least mutable PPIs. That is, since immutable PPIs are found in nearly all environments, they are more readily observed in just one. However, another possible explanation is that, in our assay, mutable PPIs are more likely to be false positives in environment(s) in which they are identified or false

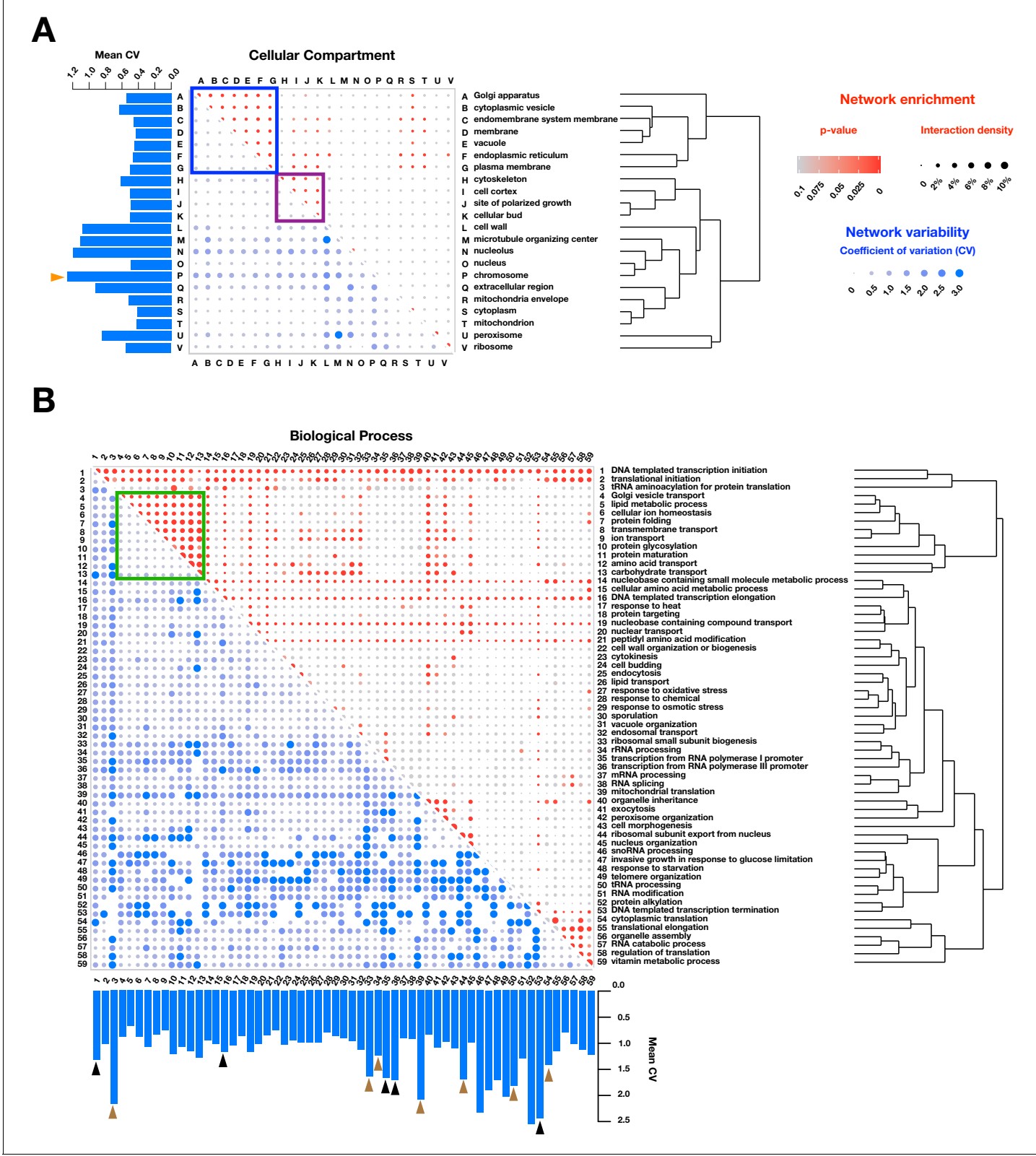

**Figure 2.** Functional enrichment of PPIs detected by PPiSeq. PPI enrichment (red) and variability (blue) across environments of gene ontology cellular compartments (A) and biological processes (B). Red node size is the percent of interacting protein pairs (interaction density) observed for a given pair of GO terms and the node color is the p-value of this percent over a random expectation. Blue node size and color are the variability (coefficient of variation, CV) of interaction densities across nine environments tested. GO terms are hierarchically clustered by the interaction density (red dots). Boxes

*Figure 2 continued on next page*

*Figure 2 continued*

mark frequently interacting and invariable cellular compartments and biological processes involved in membrane transport and protein maturation (blue and green) and cell division (purple). Barplots show the mean CV of interaction densities for each GO term across all other GO terms. Orange, black and brown triangles highlight three different groups of related GO terms: chromosome, transcription, and translation, respectively.

negatives in environments in which they are not identified. To investigate this second possibility, we first asked whether PPIs present in very few environments have lower fitnesses, as this might indicate that they are closer to our limit of detection. We found no such pattern: mean fitnesses were roughly consistent across PPIs found in 1 to 6 conditions, although they were elevated in PPIs found in 7–9 conditions (*Figure 3—figure supplement 2A*). To directly test the false-positive rate stemming from pooled growth and barcode sequencing, we validated randomly selected PPIs within each mutability bin by comparing their optical density growth trajectories against controls (*Figure 3B*). We found that mutable PPIs did indeed have lower validation rates in the environment in which they were identified, yet putative false positives were limited to ~50%, and, within a bin, do not differ between PPIs that have been previously identified and those that have been newly discovered by our assay (*Figure 3—figure supplement 2B*). We also note mutable PPIs might be more sensitive to environmental differences between our large pooled PPiSeq assays and clonal 96-well validation assays, indicating that differences in validation rates might be overstated. To test the false-negative rate, we assayed PPIs identified in only SD by PPiSeq across all other environments by optical density growth and found that PPIs can be assigned to additional environments (*Figure 3—figure supplement 2C*). However, the number of additional environments in which a PPI was detected was generally low (2.5 on average), and the interaction signal in other environments was generally weaker than in SD (*Figure 3—figure supplement 2D*). To better estimate how the number of PPIs changes with PPI mutability, we used these optical density assays to model the validation rate as a function of the mean PPiSeq fitness and the number of environments in which a PPI is detected. This accurate model (Spearman's $r = 0.98$ between predicted and observed, *Figure 3—figure supplement 3*) provided confidence scores (predicted validation rates) for each PPI (Appendix 2 Table S5) and allowed us to adjust the true positive PPI estimate in each mutability bin. Using this more conservative estimate, we still found a preponderance of mutable PPIs (*Figure 3—figure supplement 3D*). Finally, we used a pair of more conservative PPI calling procedures that either identified PPIs with a low rate of false positives across all environments (FPR <0.1%, see Materials and methods, Appendix 2 Table S6) or removed data from the most extreme environmental perturbation (16°C). Using these higher confidence sets, we still found that mutable PPIs far outnumbered others in the multi-condition PPI network (*Figure 3—figure supplement 4*).

We next examined if mutable PPIs can be distinguished from less mutable PPIs in the PPI network. We first asked whether proteins that participate in less or more mutable PPIs differed in their connectivity (degree distribution) and found that proteins from less mutable PPIs are more likely to be and interact with hubs in the multi-environment PPI network (*Figure 3—figure supplement 5A and B*). One possible explanation for these findings is that proteins in less mutable PPIs form a 'backbone' in the PPI network and that proteins in more mutable PPIs decorate this backbone. Alternatively, proteins that participate in less and more mutable PPIs may be forming distinct modules in the protein interactome, with modules of less mutable PPIs being more highly connected. To distinguish between these two possibilities, we calculated a mutability score for each protein based on the variability in fitness measures across environments for PPIs in which it participates, and compared the mutability score of each protein against the mean mutability score of its neighbors (Materials and methods, *Figure 3—figure supplement 5C and D*). We found that the neighbors of proteins in less mutable PPIs tend to be in other PPIs with low mutability, suggesting that less mutable PPIs are forming tight 'core' modules in the PPI network and that more mutable PPIs form a distinct 'accessory' module. To verify this, we applied three network community detection algorithms to our multi-environment PPI network (*Clauset et al., 2004*; *Pons and Latapy, 2005*; *Rosvall et al., 2009*). Each found three major communities that differed by the mean mutability score of their constituents: two core modules composed with a greater proportion of highly connected (hub) proteins with low or intermediate mutability scores, and an accessory module composed of less connected proteins with high mutability scores (*Figure 3C–E* and *Figure 3—figure supplement 6*). Using gene

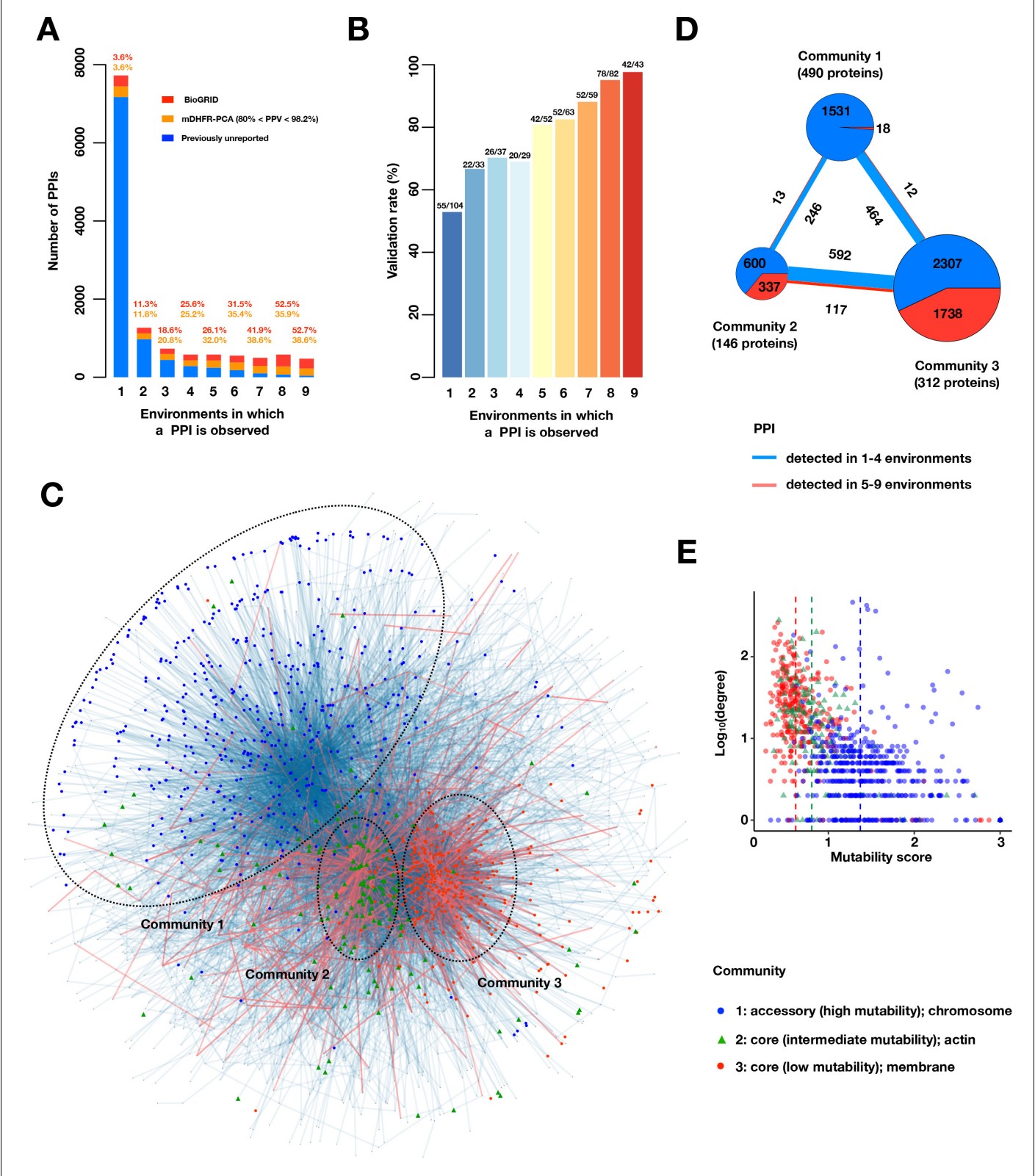

**Figure 3.** A large accessory protein interactome. (**A**) Barplot of PPI number binned by the number of environments in which a PPI is observed. Colors indicate PPIs called by both PPiSeq and BioGRID inclusive of mDHFR-PCA (red), PPIs called by PPiSeq that scored high but were not called by mDHFR-PCA (yellow), and PPIs called by PPiSeq that scored low by mDHFR-PCA (blue). (**B**) Validation rates of PPIs binned by the number of environments in which a PPI is observed. Validations were performed using OD600 trajectories of clones grown in multi-well plates. (**C**) Mutable and less mutable PPIs

*Figure 3 continued on next page*

*Figure 3 continued*

form distinct modules in the network. PPIs that are detected in at least five environments (red edges) form two tight core modules. PPIs that are detected in fewer than five environments (blue edges) form a less connected accessory module. Proteins in different modules are labeled with different shapes and colors. The network uses an edge-weighted spring embedded layout. (D) Number of PPIs within and between each community. PPIs detected in at least or fewer than five environments are shown in red and blue, respectively. The size of the square or circle is proportional to the number of PPIs. The number below each community is the number of proteins within each community. (E) Scatter plot of degrees and mutability scores of proteins in each community.

The online version of this article includes the following figure supplement(s) for figure 3:

**Figure supplement 1.** PPIs across conditions.
**Figure supplement 2.** Validating PPIs.
**Figure supplement 3.** Predicting validation rates.
**Figure supplement 4.** Mutable PPIs outnumber immutable PPIs in higher confidence PPI networks.
**Figure supplement 5.** PPIs with a similar mutability are more likely to be connected.
**Figure supplement 6.** The multi-environment PPI network contains three major communities with different mutability scores.

ontology term enrichment analysis, we found that proteins in each module were enriched for different cellular compartments. The less mutable core modules are associated with membranous compartments (low mutability) and the actin cytoskeleton (intermediate mutability), while the highly mutable accessory module is associated with the chromosome (*Figure 3E* and Appendix 2 Table S7).

## Properties of mutable PPIs

Previous work has used patterns of mRNA co-expression to predict which hub proteins are likely to participate in PPIs that are mutable across environments (calling these proteins 'date hubs'), and found that they have several distinguishing features (*Han et al., 2004*). However, the robustness of these conclusions has been vigorously debated (*Agarwal et al., 2010*; *Batada et al., 2006*; *Batada et al., 2007*; *Bertin et al., 2007*; *Yu et al., 2008*). Here, because we have a direct measure of the mutability of each PPI across environments, we are able to confidently validate or reject predictions made from co-expression data, in addition to testing new hypotheses.

We first asked whether co-expression is indeed a predictor of PPI mutability and found that it is: co-expression mutual rank (which is inversely proportional to co-expression across thousands of microarray experiments) declined with PPI mutability (*Figure 4A* and *Figure 4—figure supplement 1*; *Obayashi and Kinoshita, 2009*; *Obayashi et al., 2019*). Confirming a second prediction from co-expression data, we found that proteins from less mutable PPIs are more likely to co-localize to the same cellular compartment in standard laboratory conditions than those from more mutable PPIs (*Chong et al., 2015*), suggesting localization changes drive some changes (*Figure 4B*; *Levy et al., 2014*; *Rochette et al., 2014*). Both the co-expression and co-localization patterns were also apparent in our higher confidence PPI sets (*Figure 4—figure supplements 2A, B*, *3A and B*), indicating that they are not caused by different false positive rates between the mutability bins.

We next asked what features from existing genome-wide data sets correlate with proteins that are involved in less or more mutable PPIs. We binned proteins by their PPI degree, and, within each bin, determined the correlation between the mutability score and another gene feature (*Figure 4C* and *Figure 4—figure supplement 4*, Appendix 2 Table S8) (*Costanzo et al., 2016*; *Finn et al., 2014*; *Gavin et al., 2006*; *Holstege et al., 1998*; *Krogan et al., 2006*; *Levy and Siegal, 2008*; *Myers et al., 2006*; *Newman et al., 2006*; *Ostlund et al., 2010*; *Rice et al., 2000*; *Stark et al., 2011*; *Wapinski et al., 2007*; *Ward et al., 2004*; *Yang, 2007*; *Yu et al., 2008*). These correlations were also calculated using our higher confidence PPI sets, confirming results from the full data set (*Figure 4—figure supplements 2C, D*, *3C and D*). We found that mutable hubs (>15 PPIs) have more genetic interactions, in agreement with predictions from co-expression data (*Bertin et al., 2007*; *Han et al., 2004*), and that their deletion tends to cause larger fitness defects. However, these two correlations were weaker or not seen with non-hub proteins. Contradicting predictions from co-expression data, we did not find that mutable hubs are more quickly evolving, as measured by dN/dS (*Costanzo et al., 2016*; *Yang, 2007*), but that non-hub proteins (the vast majority of mutable PPIs) are. In addition, we found that proteins that participate in mutable PPIs tend to have lower

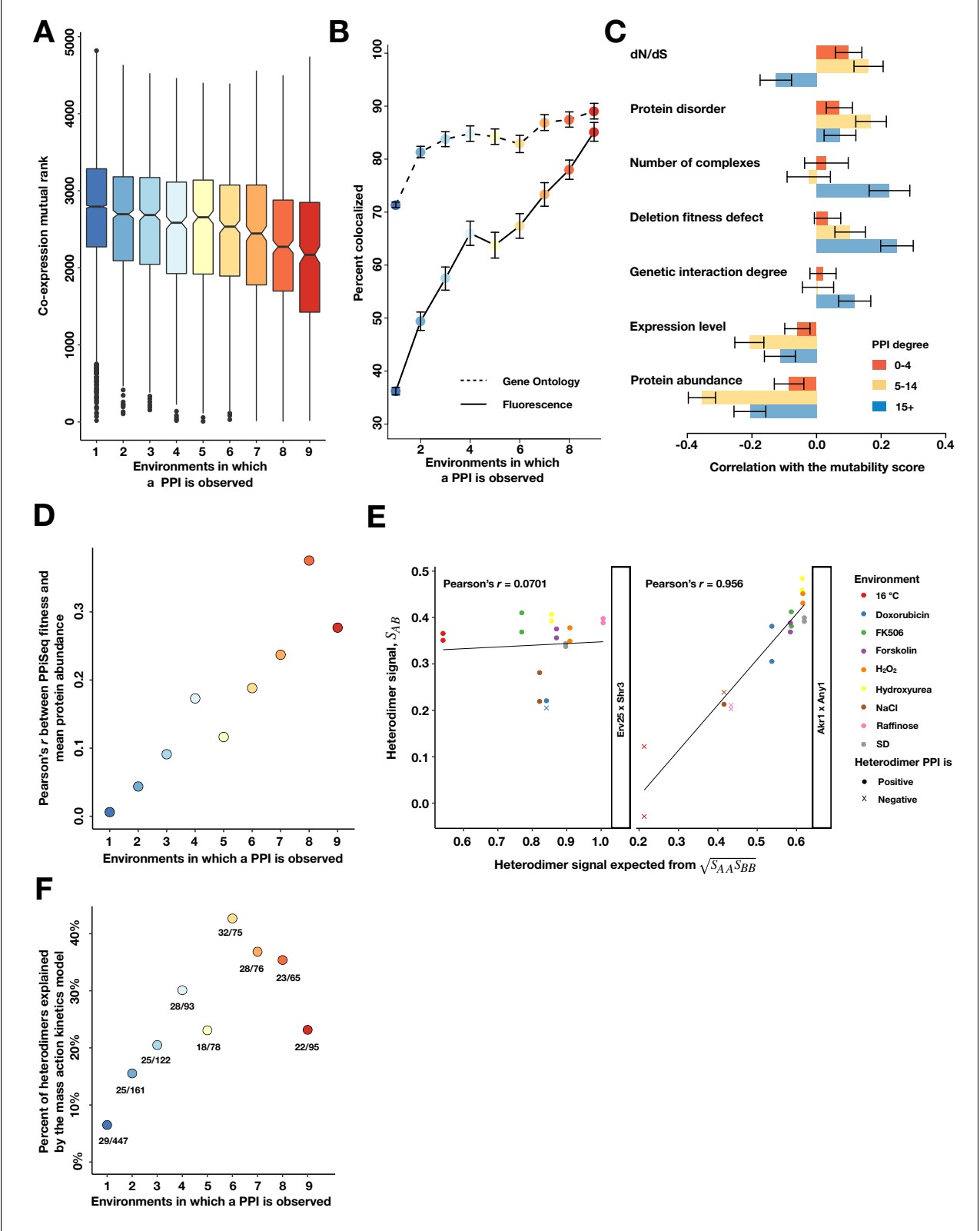

**Figure 4.** Properties of mutable and less mutable PPIs. (**A**) The co-expression mutual rank for PPIs binned by the number of environments in which the PPI is detected. A higher mutual rank means worse co-expression. Notches are the 95% confidence interval for the median, hinges correspond to the first and third quartiles, and whiskers extend 1.5 times the interquartile range. (**B**) The percent of protein pairs that have been found colocalized by gene ontology (GO Slim, dashed line) and fluorescence (solid line) (*Chong et al., 2015*). (**C**) Spearman correlation between the PPI mutability score and

*Figure 4 continued on next page*

*Figure 4 continued*

other gene features, binned a gene's PPI degree. In (B) and (C), the error bars are the standard deviation from 1000 bootstrapped data sets. (D) Pearson correlation between a PPI's fitness and geometric mean abundance of two interacting proteins in *Ho et al., 2018*, binned by the number of environments in which a PPI is detected. (E) Examples of non-significant (Erv25 x Shr3) and significant (Akr1 x Any1) predictions. Observed heterodimer fitness ($S_{AB}$) is plotted against the expectation based on the geometric mean of the two constituent homodimer fitnesses ($S_{AA}$ and $S_{BB}$). (F) Percent of heterodimers whose fitness changes can be significantly predicted by the geometric mean of the two constituent homodimers, binned by the number of environments in which a PPI is observed.

The online version of this article includes the following figure supplement(s) for figure 4:

**Figure supplement 1.** The co-expression mutual rank for PPIs detected in each condition binned by the number of environments in which the PPI is detected.

**Figure supplement 2.** Mutable PPIs and their properties for higher confidence PPI calls.

**Figure supplement 3.** Mutable PPIs and their properties, excluding the 16°C condition.

**Figure supplement 4.** Spearman correlation between the protein's mutability score and other gene features.

**Figure supplement 5.** Proteins that participate in multiple complexes are distributed over a wide range of complexes.

**Figure supplement 6.** Exploring the relationship of protein abundance, PPI abundance, and PPI mutability.

mRNA and protein expression levels than those in less mutable PPIs (*Holstege et al., 1998*; *Newman et al., 2006*), perhaps because highly abundant proteins are more difficult to post-translationally regulate. As might be expected, we also found that mutable hubs, but not non-hubs, are more likely to participate in multiple protein complexes than less mutable proteins (*Figure 4—figure supplement 5*; *Costanzo et al., 2016*). Finally, confirming another prediction from co-expression data (*Singh et al., 2007*), we found that proteins in mutable PPIs are more likely to contain intrinsically disordered regions, suggesting that they may adopt different conformations in some environments or intracellular contexts to promote a PPI. Taken together, these correlations largely confirm predictions from co-expression data, but highlight differences between hub and non-hub proteins in the core and accessory PPI networks.

Given the above results, we suspected that changes in mutable PPIs may be more likely to be driven by post-translational regulation than protein abundance changes (*Taylor et al., 2009*). We first tested if the relationship between protein abundance and PPI abundance changes with PPI mutability. For each PPI, we compared our estimated PPI abundance in standard rich media (fitness in SD) to a meta-analysis estimate of mean protein abundance of each protein pair (*Ho et al., 2018*). These two measures correlate weakly (Pearson's $r = 0.16$, *Figure 4—figure supplement 6A*). However, when binned by PPI mutability, we find that less mutable PPIs are more strongly correlated than highly mutable PPIs (*Figure 4D*). Second, we asked if quantitative changes in PPI abundance across conditions are better predicted by changes in protein abundance for less mutable PPIs. Using homodimer PPI abundance estimates (fitnesses) as a proxy for protein abundance (*Stynen et al., 2018*) in a mass-action kinetics model (Materials and methods), we used linear regression to test if changes in the estimated abundance of a set of PPIs (1212 heterodimers) are explained by our proxy measure of the constituent protein abundances (180 homodimers). We find that some heterodimers fit this simple model well (*Figure 4E*, right), while others fit poorly (*Figure 4E*, left), with 19% of heterodimers explained by the model (*Figure 4—figure supplement 6B*, Materials and methods). We next used logistic regression to determine what features may underlie a good or poor fit to the model (*Figure 4—figure supplement 6C*) and found that PPI mutability was the best predictor, with more mutable PPIs being less frequently explained (*Figure 4F*). Unexpectedly, mean protein abundance was the second best predictor, with high abundance predicting a poor fit to the model, particularly for less mutable PPIs (*Figure 4—figure supplement 6D and E*). Taken together, these data suggest that mutable PPIs are subject to more post-translational regulation across environments and that high basal protein abundance may saturate the binding sites of their partners, limiting the ability of gene expression changes to regulate PPIs.

## Rewiring of the glucose transporter network

We have shown above that most protein network rewiring occurs within a dynamic accessory module and that changes in the module are more likely to reflect condition-specific protein localization, binding, or other mechanisms, rather than by changes in protein abundance. To explore whether

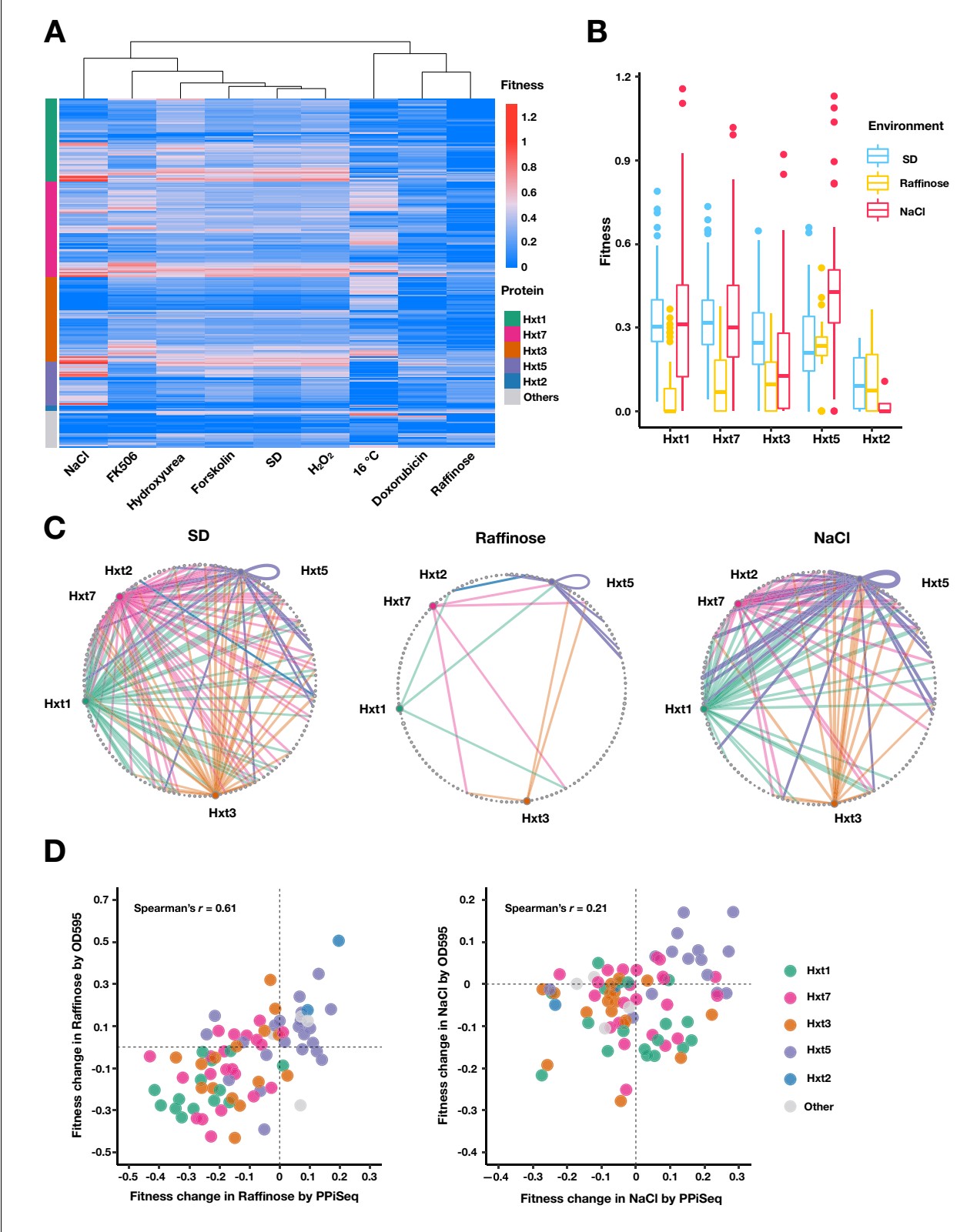

**Figure 5.** Carbohydrate transport network rewiring as captured by PPiSeq. (**A**) Heatmap of abundances (fitnesses) of PPIs involved in carbohydrate transport across different environments. (**B**) Boxplots of fitnesses of PPIs involving Hxt proteins in SD, Raffinose and NaCl environments. The bottom of each box, the line drawn in the box, and the top of the box represent the 1st, 2nd, and 3rd quartiles, respectively. The whiskers extend to ±1.5 times the interquartile range. (**C**) Circular network plots of PPIs containing Hxt proteins in SD, Raffinose, and NaCl environments. Nodes are proteins and

*Figure 5 continued on next page*

*Figure 5 continued*

colors are as in (**A**). Node size is proportional to its degree in the multi-environment PPI network. Edge width is proportional to abundance in each environment. (**D**) Scatter plot of fitness changes relative to SD as measured by PPiSeq and clonal growth dynamics for randomly chosen carbohydrate-transport PPIs in Raffinose (80 PPIs) and NaCl (90 PPIs).

coordinated changes are also occurring in the core module and what drives these changes, we examined how core PPIs involved in carbohydrate transport change across environments (*Figure 5A*). Glucose is transported into the cell using membrane-spanning hexose transporters (HXT genes) (*Boles and Hollenberg, 1997*; *Kruckeberg, 1996*; *Özcan and Johnston, 1999*). The major Hxt transporters can be divided by their extracellular glucose binding into low affinity (Hxt1, Hxt3), moderate affinity (Hxt5), and high affinity (Hxt2, Hxt4, Hxt6, Hxt7) (*Ozcan and Johnston, 1995*; *Reifenberger et al., 1995*; *Verwaal et al., 2002*), two of which were excluded from our screen (Hxt4, Hxt6). For PPIs involving at least one HXT gene product, we found that there were only minor differences in PPI abundance between most environments that contained glucose as their sugar source. However, several environments showed marked differences (NaCl, 16°C, Doxorubicin, Raffinose), presumably reflecting how transport is altered in these environments. We examined changes in the Raffinose (low glucose [*Özcan and Johnston, 1999*]) and NaCl (high glucose, osmotic stress [*Verwaal et al., 2002*]) environments more closely by plotting a carbohydrate transport sub-network in each environment (*Figure 5B and C*) and validating differences using optical density growth trajectories of 90 randomly chosen PPIs (*Figure 5D*). Most Hxt PPIs that were detectable in SD were lost in the low-glucose Raffinose environment. However, Hxt5, the only glucose transporter expressed during the stationary phase (*Verwaal et al., 2002*; *Wu et al., 2004*), maintained most of its PPIs (*Figure 5B and C*). Surprisingly, the high-affinity glucose transporter Hxt7, whose mRNA and protein expression has been reported to increase in Raffinose (*Lai et al., 2007*; *Ye et al., 2001*), lost most of its PPIs, suggesting other factors such as protein endocytosis or degradation could be important to regulating its activity over our growth cycles (*Ye et al., 2001*). In NaCl, fewer PPIs were detectable than in SD, but some PPIs, especially those involving Hxt5 increased in relative

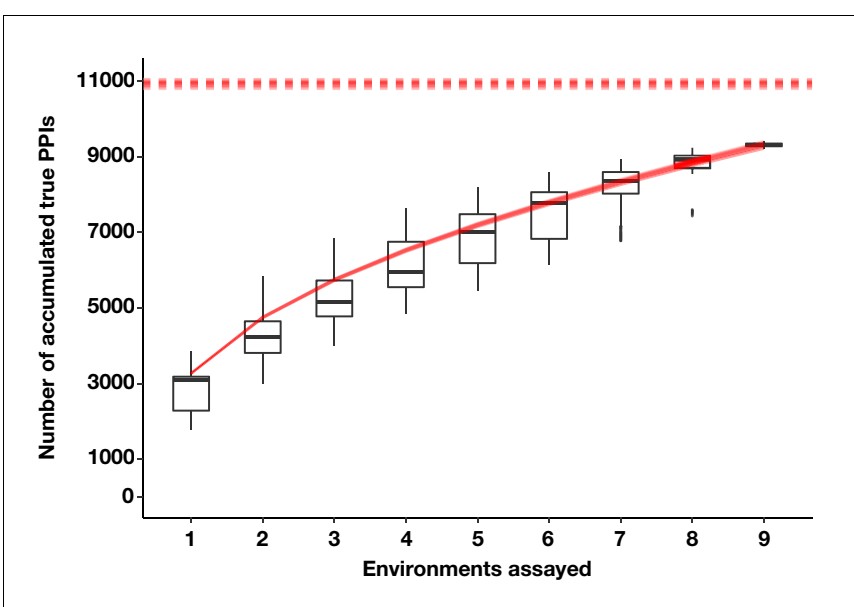

**Figure 6.** The estimated number of true PPIs discovered by PPiSeq using repeated sampling of data in permuted orders of environment addition. Boxplots summarize the distribution of the number of unique PPIs across permutations. The bottom of each box, the line drawn in the box, and the top of the box represent the 1st, 2nd, and 3rd quartiles, respectively. The whiskers extend to ±1.5 times the interquartile range. Overlayed solid red lines and dashed red lines are the Kindt exact accumulation curves and the bootstrap estimators of the total number of unique PPIs across infinite environments for each simulation, respectively.

abundance (*Figure 5B and C*). Previous work has found that HXT5 expression increases during salt stress (*Verwaal et al., 2002*), suggesting the change in Hxt5 PPIs may be a direct consequence of its change in abundance. Taken together and consistent with mechanisms of core network regulation described above, these results suggest that the changes in the glucose transport subnetwork are mainly driven by changes in protein expression (*Celaj et al., 2017*).

## Size of the pan-environment protein interactome

The yeast protein interactome has been previously estimated to contain 37,600 to 75,500 detectable interactions in standard growth conditions (*Hart et al., 2006*; *Sambourg and Thierry-Mieg, 2010*). We have shown here that much of the PPI network is rewired across conditions (*Figure 3A*), suggesting that the pan-environment PPI network is likely to be larger than these projections. To estimate how many PPIs remain to be discovered within our search space (~9% of protein pairs), we constructed several hypothetical PPI-by-environment observation matrices by randomly assigning a PPI observation at a rate proportional to its confidence score within an environment (Materials and methods, Appendix 2 Table S5). We then plotted PPI accumulation curves across permuted orders of environment addition (*Oksanen et al., 2019*) and found that the number of observed PPIs is beginning to approach saturation, with fewer PPIs accumulating with the addition of each new environment (*Figure 6*). Borrowing a species richness estimator from ecology (*Oksanen et al., 2019*), we estimate that there are ~10,840 *true* interactions within our search space across all environments, ~3 fold more than are detected in SD (note difference to *Figure 3*, which counts *observed* PPIs). This analysis shows that the number of PPIs present across all environments is much larger than the number observed in a single condition, but that it is feasible to discover most of these new PPIs by sampling a limited number of conditions.

## Discussion

We developed a massively parallel quantitative PPI assay to characterize how the yeast protein interactome changes across conditions. We found a large, previously undersampled accessory protein interactome that changes across conditions and is enriched for proteins involved in transcription, RNA processing, and translation. Mutable PPIs are less likely to co-express, co-localize, and be explained by simple mass action kinetics, and more likely to contain intrinsically disordered regions, evolve quickly, and be of low abundance in standard conditions. Taken together, these results suggest that major rewiring within the protein interactome is driven to a larger extent by post-translational regulation, with protein abundance changes being more likely to tune levels of relatively immutable PPIs in the core interactome.

Results presented here and elsewhere (*Huttlin et al., 2020*) suggest that PPIs discovered under a single condition or cell type are a small subset of the full protein interactome emergent from a genome. We sampled nine diverse environments and found approximately threefold more interactions than in a single environment. However, the discovery of new PPIs began to saturate, indicating that most condition-specific PPIs can be captured in a limited number of conditions. Testing in many more conditions and with PPI assays orthogonal to PPiSeq will undoubtedly identify new PPIs, however a more important outcome could be the identification of coordinated network changes across conditions. Using a test set of ~1.6 million (of ~18 million) protein pairs across nine environments, we find that specific parts of the protein interactome are relatively stable (core modules) while others frequently change across environments (accessory modules). However, two important caveats of our study must be recognized before extrapolating these results to the entire protein interactome across all environment space. First, we tested for interactions between a biased set of proteins that have previously been found to participate in at least one PPI as measured by mDHFR-PCA under standard growth conditions (*Tarassov et al., 2008*). Thus, proteins that are not expressed under standard growth conditions are excluded from our study, as are PPIs that are not detectable by mDHFR-PCA or PPiSeq. It is possible that a comprehensive screen using multiple orthogonal PPI assays would alter our observations related to the relative dynamics of different regions of the protein interactome and the features of mutable and immutable PPIs. Second, we tested a limited number of environmental perturbations under similar growth conditions (batch liquid growth). It is possible that more extreme environmental shifts (e.g. growth as a colony, anareobic growth, pseudohyphal growth) would introduce new accessory modules or alter the mutability of the PPIs we detect. Nevertheless,

results presented here provide a new mechanistic view of how the cell changes in response to environmental challenges, building on the previous work that describes coordinated responses in the transcriptome (*Brauer et al., 2008*; *Gasch et al., 2000*) and proteome (*Breker et al., 2013*; *Chong et al., 2015*).

We have shown that our iSeq platform is capable of building libraries that exceed one billion interactions (*Liu et al., 2019*; *Schlecht et al., 2017*), making it feasible to expand assay coverage to an entire protein interactome. While we used reconstruction of C-terminal-attached mDHFR fragments as a reporter for PPI abundance, similar massively parallel assays could be constructed with different PCA reporters or tagging configurations to validate our observations and overcome false negatives that are specific to our reporter. Indeed, the recent development of 'swap tag' libraries, where new markers can be inserted C- or N-terminal to most genes (*Weill et al., 2018*; *Yofe et al., 2016*), in combination with our iSeq double barcoder collection (*Liu et al., 2019*), makes extension of our approach eminently feasible.

Our assay detected subtle fitness differences across environments (*Figure 3—figure supplement 1B and C*), which we used as a rough estimate for changes in relative PPI abundance. While it would be tempting to use fitness as a direct readout of *absolute* PPI abundance within a cell, non-linearities between fitness and PPI abundance may be common and PPI dependent. For example, the relative contribution of a reconstructed mDHFR molecule to fitness might diminish at high PPI abundances (saturation effects) and fitness differences between PPIs may be caused, in part, by differences in how accessible a reconstructed mDHFR molecule is to substrate. In addition, environmental shifts might impact cell growth rate, initiate a stress response, or result in other unpredictable cell effects that impact the selective pressure of methotrexate and thereby fitness (*Figure 1—figure supplements 2*, *3* and *5*). Finally, our assays were performed across cycles of batch growth meaning that changes in PPI abundance across a growth cycle (e.g. lag, exponential growth, saturation) are coarse grained into one measurement. While this method potentially increases our chance of discovering a diverse set of PPIs, it might have an unpredictable impact on the relationship between fitness and PPI abundance (*Li et al., 2018*). To overcome these issues, strains containing natural or synthetic PPIs with known abundances and intracellular localizations could be spiked into cell pools to calibrate the relationship between fitness and PPI abundance in each environment. In addition, continuous culturing systems may be useful for refining precision of growth-based assays such as ours.

PPIs have often been reported as a qualitative presence or absence of a PPI, but the propensity of two proteins expressed in the same cell to form a complex is a continuum. For some analyses in this work, we use qualitative statistical calls to identify 'positive' PPIs, but our approach preserves and analyzes a quantitative signal for each PPI. Quantitative assays presented here and elsewhere (*Celaj et al., 2017*; *Diss and Lehner, 2018*; *Rochette et al., 2014*; *Schlecht et al., 2012*; *Schlecht et al., 2017*) hold promise to shift the paradigm of high-throughput PPI assays from PPI detection to *in vivo* PPI characterization. This will require novel analyses to re-conceptualize the PPI network as a continuum of interaction probabilities that are dependent not only on changes in protein abundance, but also on post-translational modifications, intracellular localization, steric effects, and competitive binding (*Stynen et al., 2018*). One important goal would be to estimate an *in vivo* 'functional binding affinity' for each PPI — an important analog to *in vitro* binding affinity that reports how PPI abundance scales with the abundance of its constituent proteins (*Kastritis and Bonvin, 2013*). Here, we use homodimer abundance as a proxy for protein abundance. However, genome-wide mRNA abundance measures could be used as a proxy for protein abundance or protein abundance could be measured directly in the same pool (*Levy et al., 2014*) by, for example, attaching a full length mDHFR to each gene using 'swap tag' libraries mentioned above (*Weill et al., 2018*; *Yofe et al., 2016*). Changes in the functional affinity across environments would point to other mechanisms of *in vivo* regulation that could be dissected in high-throughput by combining PPiSeq with mutagenesis of interacting proteins or trans-acting factors (*Diss and Lehner, 2018*).

# Materials and methods

## PPiSeq library construction
### Construction of diploid PPiSeq library

A interactome-scale protein-fragment complementation assay (PCA) screen (*Tarassov et al., 2008*), found 2770 PPIs with a positive predictive value (PPV) of 98.2%. In this study, *MATa* bait (F[1,2], 1757 strains) and *MATα* prey (F[3], 1135 strains) PCA strains that participated in a PPI with > 80% PPV (~10,000 PPIs) in *Tarassov et al., 2008* were selected for barcoding. Each strain was barcoded in duplicate, pooled with strains of the same mating type, and mated to opposite mating type strains as part of a pool to generate ~4 double barcoded strains per PPI. Strains lost at each stage of this process are detailed in *Supplementary file 1*. The final diploid PPiSeq pool contained ~6 million double barcodes, representing ~1.6 million protein pairs. Controls added to this pool are discussed later.

### Construction of haploid PPiseq libraries

Haploid PCA strains were picked from Yeast-Interactome Collection (Dharmacon, YSC5849) using the ROTOR HDA (SINGER instruments) and mated as arrays to the double barcoder collection on YPD agar plates (*Liu et al., 2019*). Following a 24 hr incubation at 30°C, diploids were selected by replicating colonies onto YPD + Nat + G418 plates (bait-F[1,2]) or YPD + Hyg + G418 plates (prey-F [3]) and incubating at 30°C for 48 hr. Selected diploids were replicated onto the sporulation plates (*Baryshnikova et al., 2010*), sealed with parafilm, and incubated at room temperature for a week. Sporulated bait-F[1,2] diploids were replicated onto SC - Met - Lys - Cys - Arg - His + Canavanine + G418 + Nat plates and incubated at 30°C for 96 hr to select *MATa* barcoded bait (F[1,2]) strains. Sporulated prey-F[3] diploids were replicated onto SC - Met - Lys - Cys - Arg - Leu + Canavanine + G418 + Hyg plates and incubated at 30°C for a week to select for *MATα* barcoded prey (F[3]) strains. To further purify barcoded haploids, they were replicated again onto the same selection plates and incubated at 30°C for another 48 hr. Purified haploids were stored at -80°C in 384-well plates.

### Generation of the diploid PPiSeq library by bulk mating

Frozen barcoded F[1,2] and F[3] strains were thawed and replicated into 384-well plates with SC - Met - Lys - Cys - Arg - His + Canavanine + G418 + Nat and SC - Met - Lys - Cys - Arg - Leu + Canavanine + G418 + Hyg liquid media, respectively. The cells were grown to saturation at 30°C for 96 hr. All F[1,2] and F[3] clones were pooled by pipetting, resulting in $2.13 \times 10^{10}$ and $3.36 \times 10^{10}$ cells, respectively. F[1,2] and F[3] pools were each transferred to independent flasks of 1 L YPD + G418 liquid media and grown at 30°C for 24 hr. The two cell pools were mixed (2 L, $5.36 \times 10^{11}$ cells), pelleted, resuspended in 50 mL water, plated onto 46 YPD plates at a density of ~$1.15 \times 10^{10}$ cells per plate, and incubated for 24 hr at 30°C to mate. All cells were scraped from YPD plates and pooled in ~250 mL of water. The number of cells in the pool was counted ($7 \times 10^{11}$ cells) and all cells were plated onto 265 YPD + Nat + Hyg + G418 plates at equal cell densities (~$2.5 \times 10^9$ cells per plate). Cells were incubated at 30°C for 48-h and then replica plated onto another 265 YPD + Nat + Hyg + G418 plates. After another 48 hr incubation at 30°C, cells were scraped from the 265 plates and pooled in ~1.3 L of water. All cells (~$1.67 \times 10^{12}$) were spun down at 1500 g, resuspended with 8.5 L YP + 2% galactose liquid media, and grown at 30°C for 48 hr. Cells were counted (~$2.88 \times 10^{12}$) and ~44.4% of cells were transferred into 16 L SC-Ura liquid media, and incubated for 72 hr at 30°C. Cells were 1:10 diluted into another 16 L SC-Ura liquid media and grown for another 48-h. Finally, all the cells were collected to form the pooled diploid PPiSeq library.

## Construction of barcoded DHFR-fragment control strains
### Overview

Bait or prey PCA constructs that nonspecifically bind to other tagged proteins will result in false positive interactions calls. To screen for these promiscuous constructs (*Tarassov et al., 2008*), we constructed barcoded strains that constitutively express one of four unlinked DHFR fragments under the *TEF1* promoter at the *HO* locus: linker-F[1,2], F[1,2], linker-F[3], and F[3], where the linker codes for a 10 amino acid (Gly.Gly.Gly.Gly.Ser)$_2$ flexible polypeptide. These strains were mated to the haploid PPiSeq libraries to generate diploid DHFR-fragment control (ORF x Null) strains. ORF x Null control

strains were spiked into the final diploid PPiSeq library. Identification and removal of promiscuous bait or prey constructs because of interactions with DHFR-fragment controls is described in Appendix 1.

## Construction and barcoding haploid strains that express DHFR fragments

The linker-DHFR[1,2]-NatMx and DHFR[1,2]-NatMX fragments were cloned from the plasmid pAG25 linker-DHFR[1,2]-NatMX (*Tarassov et al., 2008*) with primers oSL368 and oSL373, and oSL369 and oSL373, respectively (*Supplementary file 2*). Similarly, the linker-DHFR[3]-HygMx and DHFR[3]-HygMX fragments were cloned from the plasmid pAG32 linker-DHFR[3]-HygMX (*Tarassov et al., 2008*) with primers oSL368 and oSL373, and oSL370 and oSL373, respectively. These primers add BamHI and XhoI restriction sites on either end of each amplicon. BamHI and XhoI restriction sites were used to subclone amplicons downstream of the TEF1pr sequence in the pS413 TEF1pr-His3 plasmid. These plasmids were used as a PCR template to construct homologous recombination cassettes for integration at the *HO* locus. PCR was carried out using primers oSL378 and oSL379, which add 40-nucleotides of homology to the *HO* locus to either end of the aplicons. Amplicons were integrated into the *HO* locus of BY4741 (*MATα, his3Δ1, leu2Δ0, met15Δ0, ura3Δ0*) or BY4742 (*MATα, his3Δ1, leu2Δ0, lys2Δ0, ura3Δ0*) using standard lithium acetate transformation and selected on YPD + Nat or YPD + Hyg, respectively. Each of these strains was barcoded twice, as described above, resulting in strains ySL235-241 (*Supplementary file 3*).

## Bulk mating of barcoded DHFR-fragment control strains with haploid PPiSeq libraries

Pooled F[1,2] and F[3] haploid PPiSeq libraries were constructed as described above. Approximately $1.5 \times 10^9$ cells of the pooled *MATa* PPiSeq library (~$2\times10^{10}$ cells) and ~$1.5\times10^9$ cells of the pooled *MATα* PPiseq library (~$3\times10^{10}$ cells) were inoculated into 75 mL YPD + Nat + G418 and 75 mL YPD + Hyg + G418 liquid media, respectively, and grown for 16 hr. Meanwhile, barcoded *MATa* and *MATα* fragment control strains were each inoculated into 3 mL YPD + Nat + G418 or YPD + Hyg + G418 liquid media, respectively, and grown for 24 hr. 1 mL of each of the four barcoded *MATa* fragment control strains (ySL235-238) was added to 80 mL YPD + Nat + G418 liquid media and grown for 16 hr. Similarly, 1 mL of each of the four barcoded *MATα* fragment control strains (ySL239-242) was added to 80 mL YPD + Hyg + G418 liquid media and grown for 16 hr. These fragment control strain libraries were each mixed with the corresponding haploid PPiSeq library at equal cell numbers to yield mating pools of $2 \times 10^{10}$ cells. Mating pools were pelleted, resuspended in water and plated on two YPD plates at a density of $10^{10}$ cells/plate. Cells on mating plates were incubated for 24 hr at 30°C, scraped and pooled in water (~$3.6\times10^{10}$ cells). Cells were plated onto 10 YPD + Nat + Hyg + G418 plates at equal cell densities, incubated for 48 hr at 30°C, and replica plated onto another 10 YPD + Nat + Hyg + G418 plates. After another 48 hr incubation at 30°C, cells were scraped, pooled in water, counted (~$6\times10^{10}$ for each mating), spun down, transferred to 350 mL YP + Galactose liquid media, and grown for 24 hr at 30°C. Next, 120 mL cells from each mated pool were transferred into 1 L SC - Ura liquid media and incubated for 48 hr at 30°C. Cells from each pool were diluted 1:10 into another 1 L of SC - Ura liquid media and grown for another 48 hr. The two polls were mixed at a ratio such that the final pool contains ~2000 copies of each genotype.

### Adding spike-in controls to the diploid PPiSeq library

We spiked-in several different barcoded control strains into the diploid PPiSeq library at various frequencies to aid in the downstream data analysis (*Supplementary file 4*). In addition to DHFR-fragment controls developed here, several other previously developed controls were spiked-in: 10 positive control strains containing a full length of mDHFR (DHFR+), 100 negative control strains lacking a mDHFR (DHFR-), a set of 70 likely protein-protein interaction pairs (positive reference set or PRS) (*Liu et al., 2019*; *Yu et al., 2008*), and a set of 67 random pairs (random reference set or RRS) (*Liu et al., 2019*; *Yu et al., 2008*). After mixing, the pool of $1.5 \times 10^{11}$ cells was centrifuged, washed with SC-Ura liquid media, transferred into 6 L SC-Ura liquid media, and grown at 30°C for 24 hr to form the final PPiSeq library.

## Cell growth

The final PPiSeq library was grown in serial batch culture in 14 environments with 0.5 µg/mL methotrexate, 9 of which were used for PPI detection (*Supplementary file 5*, see below for a discussion why some environments could not be used). The PPiSeq library was grown in the SD environment twice, with replicates being performed in different labs and at different times (*Figure 1F*). For each environment, $\sim$7.4$\times$10$^9$ of frozen cells were inoculated into 1.2 L media in a 2 L Delong flask (Bellco). The cells were grown for a total of 21 generations, bottlenecking 1:8 every 48 hr ($\sim$3 generations between transfers). Bottlenecks were performed by pelleting 150 mL of the evolution and then transferring the pellet into 1.2 L of fresh media. At each bottleneck, the cell concentration was measured to estimate the number of generations that passed. The $\sim$1.05 L of cells remaining after bottlenecking was centrifuged, resuspended in water ($\sim$1$\times$10$^9$ cells/mL) and then stored in $-20$ °C for downstream processing.

The methotrexate concentration chosen (0.5 µg/mL) was previously determined to cause a $\sim$30% fitness deficit for cells that lack DHFR in SD (*Schlecht et al., 2017*). Prior to barcode sequencing, we assessed whether this methotrexate concentration causes a similar fitness deficit in the other environments by monitoring growth of cells that lack DHFR (DHFR- control stains) in that environment with or without the addition of methotrexate (*Figure 1—figure supplement 5*). To our surprise, the fitness cost of 0.5 µg/mL methotrexate varied considerably between environments, with some environments showing similar growth curves of the DHFR- cells in the presence and absence of methotrexate. Because our ability to detect PPIs depends on methotrexate exerting a large fitness cost in the absence of any DHFR molecules, five environments with small fitness costs were not processed further. In future studies, we recommend first tuning the methotrexate concentration in each environment so that all environments cause a similar fitness deficit.

## Barcode sequencing

Genomic DNA was extracted using the MasterPure Yeast DNA purification Kit (epicentre) with modifications, as described (*Schlecht et al., 2017*). Double barcodes were then amplified using a two-step PCR protocol (*Levy et al., 2015*; *Liu et al., 2019*). First, a four-cycle PCR was performed with OneTaq polymerase (New England Biolabs) in 200 reactions (125 µL/reaction), with 500 ng of genomic DNA template per reaction ($\sim$500 copies per double barcode total). Primers for this reaction follow this format:

> Forward:
> ACACTCTTTCCCTACACGACGCTCTTCCGATCTNNNNNNNNNXXXXXTTAATATGGAC
> TAAAGGAGGCTTTT
> Reverse:
> CTCGGCATTCCTGCTGAACCGCTCTTCCGATCTNNNNNNNNNXXXXXXXXXXTCGAATTCAAGC
> TTAGATCTGATA.

The Ns in these sequences correspond to any random nucleotide and are used as Unique Molecular Identifiers (UMIs) in the downstream analysis to remove skew in the counts caused by PCR jackpotting. The Xs correspond to a one of several multiplexing tags, which allows different samples to be distinguished when loaded on the same sequencing flow cell. The complete list of all primers used here are in Appendix 2 Table S3 of *Liu et al., 2019*. The PCR products were pooled and purified with NucleoSpin Gel and PCR Clean-up columns (Macherey-Nagel) at 20 reactions per column. A second 22-cycle PCR was performed with PrimeSTAR HS polymerase (Takara) in 20 X125 µL reactions. For each reaction, 30 µL of purified product from the previous PCR was used as template, and primers were standard Illumina paired-end ligation primers (pE1 and pE2 [*Liu et al., 2019*]). PCR products were pooled and purified by NucleoSpin Gel and PCR Clean-up columns (Macherey-Nagel) at five reactions per column. Amplicons were further purified by agarose gel electrophoresis. Because barcode amplicons may form heteroduplex DNA, with a different electrophoretic mobility, the complex PCR product typically formed a smear with apparent sizes between 400 bp and 3 kb. To prevent biases that may be caused by differences between barcodes in the propensity to form heteroduplex molecules, we cut out an agarose chunk that spanned this entire smear and used this for sequencing. Purified amplicons were quantified by Qubit fluorometry (Life Technologies, Q32854), pooled, and pair-end sequenced on Illumina Hiseq 4000 with 25% balanced DNA spike-in.

**Barcode sequencing analysis**

Barcode counting

Barcode reads were processed with custom written software in Python, as described (*Levy et al., 2015*; *Schlecht et al., 2017*), with modifications (available in Github https://github.com/sashaflevy/PPiSeq). Briefly, sequences were first sorted by their multiplexing tags, and then parsed to isolate the two barcodes (26 base pairs each) and two UMIs (eight base pairs each). Barcode reads were removed if they failed to pass any of three quality filters: (1) The average Illumina quality score for both barcode regions must be greater than 30, (2) the first barcode must match the regular expression: '\D*?(.ACC|T.CC|TA.C|TAC.)\D{4,7}?AA\D{4,7}?TT\D{4,7}?TT\D{4,7}?(.TAA|A.AA|AT.A|ATA.)\D*', (3) the second barcode must match the regular expression: '\D*?(.ACC|T.CC|TA.C|TAC.)\D{4,7}?AA\D{4,7}?AA\D{4,7}?TT\D{4,7}?(.TAC|T.AC|TT.C|TTA.)\D*'. Each barcode (of 2) was independently clustered with Bartender (default settings except z = −1) (*Zhao et al., 2018*). Bartender cluster centroids (with associated read indices) were matched to the known barcode sequences (*Liu et al., 2019*) if they were less than two mismatches away (Levenshtein distance). These matches provided a list of all read indices for a particular known barcode. The read indices for all unique double barcodes were identified, and the number of unique UMI combinations at these indices was used as the ultimate count for each double barcode. For all time points and conditions, the average double barcode read count was greater than 30.

Correction for putative PCR chimeras

PCR chimeras are two barcodes that stem from two different templates that are merged during PCR or sequencing (*Jaffe et al., 2019*; *Schlecht et al., 2017*; *Sinha et al., 2017*). PCR chimeras are most easily detected by the presence of an erroneous double barcode (that is not in the pool) made from two single barcodes that are in the pool, but paired with other barcodes. However, PCR chimeras can also inflate counts of double barcodes that are in the pool. We have previously found that the number of PCR chimeras for a barcode pair (BC1-BC2) scales linearly with the product of the total count of each constituent barcode in the pool across all double barcodes (BC1*BC2) (*Schlecht et al., 2017*). This linear relationship can vary subtlety from sample to sample. To determine the relationship for each sample, we therefore fit a line (using the lm() function in R) for each BC1-BC2 combination that did exist in the experimental pool at each time point in each environment. This line was used to estimate the expected number of PCR chimeras for each BC1-BC2 combination that did exist in the pool. The expected number of PCR chimeras was subtracted from the observed BC1-BC2 read count to generate a corrected read count.

**Fitness estimation**

Fitness of each double barcode was estimated by likelihood maximization using Fit-Seq (default settings) (*Li et al., 2018*). Some double barcodes were present in the pool at low or undetectable frequencies, making fitness estimation unreliable. Prior to performing Fit-Seq, we therefore merged all lineages that were likely to result in a poor fitness estimate into a single large lineage. If a lineage met any of the following criteria, it was merged: (1) it contains less than three times points with a read count greater than zero, (2) it has no time point with greater than four reads, or (3) it has less than 10 reads across all time points. In all environments, the merged lineage trajectory was similar to negative control lineages, indicating that merged lineages are likely to be negative for PPIs. To remove poor fitness estimates that were not initially merged, we calculated the difference (*d*) between the observed trajectory and the predicted trajectory (from Fit-Seq) given the estimated fitness:

$$ d = \sqrt{\left( \frac{Predicted\ count_i - Observed\ count_i}{Observed\ count_i} \right)^2} $$

where *i* is the generation number.

We defined lineages with poor fitness estimates as those with d >= 19 and removed them from downstream analyses.

## Identification of PPIs

In each environment and for each protein pair, we calculated a mean fitness value (*f*) and a p-value (*p*) against negative controls from its replicate fitness measurements. We called PPIs using a combination of *f* and *p* by using a dynamic threshold that rendered the best balance between precision and recall, as assessed using reference sets constructed from the BioGRID database. We removed PPIs containing 'promiscuous proteins' by identifying proteins that interact with untethered mDHFR fragment controls. Details can be found in Appendix 1.

## PPI validation by split mDHFR clonal growth dynamics

*MATa* (BAIT-DHFR-F[1,2]-NatMX) and *MATα* (PREY-DHFR-F[3]-HgyMX) PCA strains were cherry-picked from the Yeast-Interactome Collection (Dharmacon, YSC5849) and mated on YPD plates using a ROTOR HDA (SINGER instruments). A negative control diploid strain that lacks DHFR was generated by mating a *MATa* (HO::NatMX) strain with a *MATα* (HO::HygMx) strain. Following mating, cells were plated onto YPD + Nat + Hyg agar and grown for 48 hr at 30°C. Each diploid colony was re-arrayed into 3 wells of YPD + Nat + Hyg liquid media in the same 96-well plate. Negative control diploids were manually inoculated into 6 wells of each 96-well plate. Diploids (PPIs) on each plate were randomly picked using 'sample' function in R (replace = FALSE) from PPIs that meet specific requirements. Plates were grown for 24 hr at 30°C, and then stored in 15% glycerol at −80°C. Frozen cells were thawed and inoculated into a new 96-well plate filled with yeast nitrogen base + ammonium sulfate + His + Leu + Ura + glucose and grown for 48 hr at 30°C. Cells were next inoculated into a fresh 96-well plate containing the condition media and grown for 48 hr at 30°C. Growth conditions are as listed in *Supplementary file 5*, plus an additional 72 mg/L uracil, since these diploids do not contain *URA3*. For growth curve measurements, cells were re-inoculated into fresh media, and the optical density (OD600) of each culture was monitored every 15 mins for 48 hr using a GENios microplate reader (Tecan). The optical density trajectory of each strain was also measured in the absence of methotrexate (MTX-) as a control. The area under the curve (AUC) for each well was calculated as the sum of all OD readings right after saturation (40 hr for MTX + and 25 hr for MTX-). We compared the replicate AUCs of each protein pair ($AUC_{target\ MTX\ +}$ - $AUC_{target\ MTX\ -}$) against those of negative controls ($AUC_{control\ MTX\ +}$ - $AUC_{control\ MTX\ -}$) using a one-sided *Welch's t*-test, and calculated an adjusted p-value (Q-value) for each protein pair. A protein pair was considered as a validated PPI if Q-value <= 0.05. Since there is a marginal difference of AUC between positive PPIs and the negative controls in the absence of MTX, we did not measure AUCs without MTX when validating PPI dynamics across different conditions. The change in AUC for a protein pair in a specific condition compared to SD was calculated as follows:

$$AUC\ change_{condition} = \left( \frac{AUC_{target\ MTX+} - AUC_{control\ MTX+}}{AUC_{control\ MTX+}} \right)_{condition} - \left( \frac{AUC_{target\ MTX+} - AUC_{control\ MTX+}}{AUC_{control\ MTX+}} \right)_{SD}$$

## Network density within Gene Ontology terms

Gene Ontology (GO) terms of each protein were obtained from SGD (20190405, https://downloads.yeastgenome.org/curation/literature/go_slim_mapping.tab). Non-informative GO terms ('cellular_component', 'biological_process', 'molecular_function', 'not_yet_annotated', 'other') were removed. GO terms cover three domains: cellular compartment (CC), biological process (BP), and molecular function (MF). In this analysis, a PPI is considered as an interaction between two GO terms. For each GO term interaction, the interaction density was calculated as the ratio of the number of PPIs identified over the number of protein pairs assayed. To estimate GO term enrichment in our PPI network, we constructed 1000 random networks by replacing each bait or prey protein that was involved in a PPI with a randomly chosen protein from all proteins in our screen. This randomization preserves the degree distribution of the network. The interaction density for a GO term pair was calculated across all random networks and the distribution of network densities was used to assess statistical significance of the real network using a one-way Fisher-Pitman permutation test (*Hothorn et al., 2006*). The coefficient of variation (CV) of interaction density is the variability in PPI number for each GO term pair across all nine environments tested. The variability for an individual GO term was calculated by averaging the CVs for all GO term interactions in which it participates.

## Modeling validation rate

For each PPI, we obtained three measurements: the mean fitness value ($f$), the $\log_{10}$(p-value) against negative controls ($p$), and the number of environments in which the PPI is detected ($n$). In Appendix 1, we identified positive PPIs by using a combination of $f$ and $p$. In the section on PPI validation, we re-tested 502 PPIs by comparing their optical density growth trajectories against controls. With this validation data, we aimed to predict a validation rate ($v$) for each PPI based on $f$ and $n$. Each of these two features was a good predictor by itself (*Figure 3—figure supplement 3A and B*). To further improve predictions, we split the 502 re-tested PPIs into different bins of $f$ and $n$. When binning by $f$ for example, if there are 50 PPIs with $0.25 < f <= 0.26$, and 45 of them re-tested as positive, then $v_{bin}$ is 0.9 (45/50), $f_{bin}$ is the mean $f$ of the 50 PPIs, and $n_{bin}$ is the mean $n$ of the 50 PPIs. Similarly, when binning by n, we calculated $v_{bin}$ and $f_{bin}$ by taking the mean $v$ and $f$ of PPIs within that bin. By following this procedure, we accumulated two datasets (one binned on f, and the other binned on n) of how the $v_{bin}$ depends on $f_{bin}$ and $n_{bin}$. We trained a linear model ($lm(v_{bin} \sim f_{bin} + n_{bin})$ in R) with one dataset, and examined the accuracy of the model with the other dataset. The model performed well with the validated data (*Figure 3—figure supplement 3C*). We then applied this model to each PPI in each environment to calculate a predicted validation score.

## Calculating mutability scores

We obtained replicate fitness values for each PPI in each environment. Due to the differences in the range of fitness values in different environments (*Figure 1—figure supplement 5*), we first normalized fitness values for each PPI replicate (barcode) using the following formula:

$$Normalized\ Fitness_{barcode} = \frac{Fitness_{barcode} - Fitness_{DHFR(-)}}{Fitness_{DHFR(+)} - Fitness_{DHFR(-)}}$$

where $Fitness_{barcode}$ is the fitness of each double barcoded PPiSeq strain, $Fitness_{DHFR(-)}$ is the mean fitness of 100 control strains that lack a DHFR reporter, and $Fitness_{DHFR(+)}$ is the mean fitness of 10 control strains that have a full-length DHFR reporter under the *TEF1* promoter. From these replicate normalized fitness values, we calculated an average fitness value for each protein pair in each environment. For protein pairs with a different protein chimera acting the bait protein (i.e. ORF1-DHFR [1,2] X ORF2-DHFR[3] and ORF1-DHFR[3] X ORF2-DHFR[1,2]), only one version was kept and the average fitness was used. Negative mean fitness values, which are likely due to measurement error of a non-interacting protein pair, were replaced with 0. Using these normalized fitness values, we calculated the coefficient of variation (CV) for each PPI using its fitness values across all environments. The mutability score for each protein was then calculated by averaging the CVs for PPIs in which it participates. The mutability score for a target protein's neighbor was calculated as above, only the interaction between the target protein and its neighbor was first removed.

## Network visualization and community detection

The comprehensive PPI interaction network was generated with Cytoscape (v3.7.2) (*Shannon et al., 2003*) using the edge-weighted spring embedded layout with the default setting. Glucose transport related PPI networks were generated by igraph (R package) (*Csárdi and Nepusz, 2006*) using the layout_in_circle layout. Communities were identified with three algorithms: Fast-Greedy (*Clauset et al., 2004*), Walktrap (*Pons and Latapy, 2005*), InfoMAP (*Rosvall et al., 2009*), which were implemented by igraph (R package) (*Csárdi and Nepusz, 2006*) with default settings. Fast-Greedy is a bottom-up hierarchical approach, which tries to optimize modularity in a greedy manner. Initially, every vertex belongs to a separate community, and communities are merged iteratively so that each merge yields the largest increase in modularity. The algorithm stops when the maximum modularity is reached. Walktrap is an approach based on random walks, with the rationale that walks are more likely to be trapped within the same community because there are only a few edges that lead outside a community. It uses the results of these random walks to merge separate communities in a bottom-up manner like the Fast-Greedy algorithm. Fast-Greedy and Walktrap suffer from a resolution limit (small communities below a size threshold will always be merged to neighboring communities), and thus cannot detect small communities. The InfoMap algorithm uses community partitions of the graph as a Huffman code that compresses the information about a random walker exploring the network. It finds an optimal partition that assigns nodes to modules such that the information

needed to compress the movement of the random walkers is minimized. The three major communities detected by the InfoMAP algorithm (*Rosvall et al., 2009*) are shown in *Figure 3C*.

## Gene Ontology enrichment of network communities

We examined whether proteins in three major communities detected by infoMAP (*Rosvall et al., 2009*) were overrepresented in specific cellular compartments or biological processes. Gene Ontology enrichment was implemented with a conditional hypergeometric algorithm using the GOstats R package (*Falcon and Gentleman, 2007*). The set of proteins used for enrichment comparison are proteins that are involved in at least one PPI as determined by PPiSeq.

## Co-expression analysis

Co-expression mutual rank scores were downloaded from COXPRESdb v7.3 (https://coxpresdb.jp/) and used directly to determine if differences exist between PPI groups (*Obayashi et al., 2019*).

## Colocalization rate analysis

Two proteins were defined as colocalized by fluorescence if they localize to at least one cellular compartment in common in synthetic medium, as identified from high-throughput microscopy of GFP-fusion proteins (*Chong et al., 2015*). The colocalization rate for a PPI set was calculated by only considering proteins pairs for which localization is reported for both proteins. Two proteins were defined as colocalized by gene ontology if they are annotated to the same GO-slim cellular compartment annotation, not counting co-annotation to the general terms 'other', 'membrane', or 'cellular_component'. Bootstrapping was performed by sampling with replacement from all detected PPIs prior to binning PPIs (by number of environments detected) and determining the colocalization rates within those bins.

## Features of genes that participate in mutable and less mutable PPIs

Genes were binned by their PPI degree, as reported in BioGRID (*Stark et al., 2006*), and protein mutability scores were correlated with various previously defined gene features (*Costanzo et al., 2016*; *Finn et al., 2014*; *Gavin et al., 2006*; *Holstege et al., 1998*; *Krogan et al., 2006*; *Levy and Siegal, 2008*; *Myers et al., 2006*; *Newman et al., 2006*; *Ostlund et al., 2010*; *Rice et al., 2000*; *Stark et al., 2011*; *Wapinski et al., 2007*; *Ward et al., 2004*; *Yang, 2007*; *Yu et al., 2008*). Gene features are from *Costanzo et al., 2016*. Correlations binned by PPI degree are reported in *Figure 4C* and unbinned correlations are reported in *Figure 4—figure supplement 4*. Bootstrapping was performed by sampling with replacement from all genes that participate in at least one PPI prior to determining the correlation between the PPI variability score and a gene feature.

## Protein and PPI abundance analysis

Protein abundance was taken from a meta-analysis study (*Ho et al., 2018*) and compared to PPI fitness in the SD environment, under the assumption that it is the most comparable to standard lab growth conditions. The base R 'cor' function was used to calculate the Pearson correlation of the PPiSeq signal of each PPI to the geometric mean abundance of the two constituent proteins. We then repeated this analysis on sets of PPIs binned by the number of environments in which a PPI was detected.

## Homodimer/heterodimer mass-action kinetics model

To estimate how variation in protein abundance across environments affects abundance of the PPI complex, we used a simple mass-action kinetics model of two proteins, of concentrations $[A]$ and $[B]$, binding to form a dimer $[AB]$. This relationship can be expressed as $[AB] = k_f^{AB}[A][B]$, where $k_f^{AB}$ is a constant that reflects the population-average functional affinity of the two proteins. We assume that this constant also encompasses effects from heterozygosity (only 1/4 of heterodimers contain complementary mDHFR tags) and that cell volume does not change. We assume that the fitness $F$ of a PPiSeq strain depends linearly on the concentration of a dimer of complementary tagged proteins, such as $F^{AB} = k_{mDHFR}[AB]$. In order to model the expected $[AB]$, we used the fitness of the homodimers ($F^{AA}$ and $F^{BB}$) as a proxy for abundance of the constituent proteins ($[A]$ and $[B]$), assuming that most of the proteins are dissociated. Getting a relationship like $[A] \sim \sqrt{\frac{F^{AA}}{k_{mDHFR}k_f^{AA}}}$ for both

homodimers $AA$ and $BB$, we use this to model the fitness of the strain corresponding to the hetero-dimer $AB$ as $F^{AB} \sim \frac{k_f^{AB}}{\sqrt{k_f^{AA} k_f^{BB}}} \sqrt{F^{AA} F^{BB}}$. Thus, the fitness of the heterodimer strain should correspond to the geometric mean of the fitnesses of the homodimers of the constituent proteins, scaled by a term relating the functional affinities of each dimer. This model does not capture the complexity of *in vivo* cellular biology, but serves as a simple quantitative tool to dissect the contribution of these two factors (abundance and functional affinity) with respect to each other.

We selected data from all heterodimer PPIs amongst all homodimer-participating proteins, requiring that each PPI be quantified in at least four conditions and be considered positive in at least one condition. We pooled these measurements of PPIs from both tag configurations (F[1,2]-F[3] or F [3]-F[1,2]), if available, collecting a dataset of 1,212 PPIs amongst 180 proteins. With this, we used ordinary least-square linear regression in R to fit a model of the geometric mean of the homodimer signals multiplied by a free constant and plus a free intercept. Significantly explained heterodimer PPIs were judged by a significant coefficient (FDR < 0.05) of homodimer fitnesses predicting the het-erodimer fitness (slope) and an insignificant intercept (p-value>0.05, single-test). This criteria was used to identify PPIs for which protein expression does or does not appear to play as significant of a role as other post-translational mechanisms.

To select protein/PPI features that may be associated with being explained or not explained by this expression-variation mass-action model, we collected features with sufficient data from *Byrne and Wolfe, 2005*; *Cherry, 2015*; *Chong et al., 2015*; *Costanzo et al., 2016*; *Marchant et al., 2019*; *Stark et al., 2006*. We constructed a dataset from these by treating each heterodimer as a separate observation for each of the two constituent proteins, thus associating the features of a protein with the modeling result for each considered heterodimer in which it partici-pates. We then used the 'glm' function in R to fit a logistic model. We adjusted the resulting p-val-ues for each feature's coefficient by the Benjamini-Hochberg correction.

## Estimating the size of the pan-environment PPI network

To estimate the size of the pan-environment PPI network, we used a bootstrap approach from the ecology R package *vegan* (*Oksanen et al., 2019*). However, false positives will inflate this estimate, so we first sought to obtain a more confident estimate of the true PPIs detected. We took our data-set of each PPI observed in each condition, and then sub-sampled each PPI in each environment at a rate of the modeled validation rates (as calculated above, Appendix 2 Table S5). We repeated this procedure 32 times, and for each of these trials then calculated the Kindt exact accumulation curve and the bootstrapped species richness estimate, treating each PPI analogous to a species. Then for each sub-sample, we randomly permuted the order of environment addition and calculated the curve of the accumulated number of unique PPIs 10,000 times.

## Data and software availability

Raw barcode sequencing data are available from the NIH Sequence Read Archive as accession PRJNA630095 (https://trace.ncbi.nlm.nih.gov/Traces/study/?acc=SRP259652). Barcode sequences, counts, fitness values, and PPI calls are available in the Supplementary Tables (https://osf.io/jmhrb/). Additional data to make figures are available in Mendeley data (https://data.mendeley.com/data-sets/9ygwhk5cs3/2) and Open Science Framework (https://osf.io/7yt59/) as detailed in code reposi-tory README files. Analysis scripts are written in R and Python. All code used to analyze data, perform statistical analyses, and generate figures is available at Github (*Liu, 2020a*; copy archived at https://github.com/elifesciences-publications/PPiSeq).

## Acknowledgements

We are grateful to Stephen Michnick for providing the pAG25 linker-DHFR[1,2]-NatMX and pAG32 linker-DHFR[3]-HygMX plasmids, to Charlie Boone and Michael Costanzo for providing a dataset of different gene features, to Aaron Neiman for providing the pS413-TEF1pr-His3 plasmid, to Robert St. Onge and Ron Davis for providing the equipment to measure yeast growth curves, to Xiaoyu Zhao, Takeshi Matsui, Jamie Blundell, David Catoe, Adam Dziulko, Danielle Francois, and Richard Bennett for the assistance or suggestions on the experiments and data analysis, to Robert St. Onge and Guillaume Diss for suggestions on the manuscript. This work was supported by a grant from the

US National Institutes of Health (R01HG008354 to SL), the Louis and Beatrice Laufer Center, the New York State Center for Biotechnology, and the Joint Initiative for Metrology in Biology.

## Additional information

### Competing interests

Xianan Liu, Sasha F Levy: SFL and XL have filed a patent application (WO2017075529A1) on the double barcoding platform used in this manuscript. The other authors declare that no competing interests exist.

### Funding

| Funder | Grant reference number | Author |
|---|---|---|
| National Institutes of Health | R01HG008354 | Sasha F Levy |
| Louis and Beatrice Laufer Center | | Sasha F Levy |
| New York State Center for Biotechnology | | Sasha F Levy |
| Joint Initiative for Metrology in Biology | | Sasha F Levy |

The funders had no role in study design, data collection and interpretation, or the decision to submit the work for publication.

### Author contributions

Zhimin Liu, Resources, Data curation, Software, Formal analysis, Validation, Investigation, Visualization, Methodology, Writing - original draft, Writing - review and editing; Darach Miller, Formal analysis, Investigation, Visualization, Writing - original draft, Writing - review and editing; Fangfei Li, Software, Formal analysis; Xianan Liu, Methodology; Sasha F Levy, Conceptualization, Formal analysis, Supervision, Funding acquisition, Visualization, Methodology, Writing - original draft, Project administration, Writing - review and editing

### Author ORCIDs

Zhimin Liu ![ORCID] https://orcid.org/0000-0002-9333-8101
Darach Miller ![ORCID] https://orcid.org/0000-0002-2264-7900
Sasha F Levy ![ORCID] https://orcid.org/0000-0002-0923-1636

### Decision letter and Author response

Decision letter https://doi.org/10.7554/eLife.62365.sa1
Author response https://doi.org/10.7554/eLife.62365.sa2

## Additional files

### Supplementary files

- Supplementary file 1. Strain losses during barcoding and pool construction.

- Supplementary file 2. Primers used in the construction of DHFR-fragment control strains.

- Supplementary file 3. Barcoded haploid DHFR-fragment control strains.

- Supplementary file 4. Strains in the PPiSeq library.

- Supplementary file 5. Description of the environmental conditions tested. Cells were shaken at 220 rpm. In SD, 0.2% DMSO was added as a vehicle control.

- Transparent reporting form

## Data availability

Raw barcode sequencing data are available from the NIH Sequence Read Archive as accession PRJNA630095 (https://trace.ncbi.nlm.nih.gov/Traces/study/?acc=SRP259652). Barcode sequences, counts, fitness values, and PPI calls are available in the Supplementary Tables (https://osf.io/jmhrb/). Additional data to make figures are available in Mendeley data (https://data.mendeley.com/datasets/9ygwhk5cs3/2) and Open Science Framework (https://osf.io/7yt59/) as detailed in code repository README files. Analysis scripts are written in R and Python. All code used to analyze data, perform statistical analyses, and generate figures is available at Github (https://github.com/sashaflevy/PPiSeq; copy archived at https://github.com/elifesciences-publications/PPiSeq).

The following datasets were generated:

| Author(s) | Year | Dataset title | Dataset URL | Database and Identifier |
|---|---|---|---|---|
| Liu Z, Miller D, Levy S | 2020 | Protein-protein interaction network rewiring across environments | https://doi.org/10.17632/9ygwhk5cs3.2 | Mendeley Data, 10.17632/9ygwhk5cs3.2 |
| Liu Z, Miller D, Levy S | 2020 | Protein-protein interaction network dynamics across environments by DNA barcode sequencing | https://trace.ncbi.nlm.nih.gov/Traces/study/?acc=SRP259652&o=acc_s%3Aa | NIH Sequence Read Archive, PRJNA630095 |
| Liu Z | 2020 | Protein-protein interaction network rewiring across environments | https://osf.io/jmhrb/ | Open Science Framework, jmhrb |

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

## Appendix 1

### Identification of PPIs

In each environment and for each protein pair, we calculated a mean fitness value (*f*) and a p-value (*p*) against negative controls from its replicate fitness measurements. We called PPIs using a combination of *f* and *p* by using a dynamic threshold that rendered the best balance between precision and recall, as assessed using reference sets constructed from the BioGRID database. We removed PPIs containing 'promiscuous proteins' by identifying proteins that interact with untethered mDHFR fragment controls.

### Merging two replicate SD datasets

Since we collected fitness measurements from two independent growth cultures in the SD environment, we merged the data from these two experiments prior to PPI calling. Because the range of fitness values of the two SD datasets differed slightly, we first normalized the fitness value for each barcode:

$$Normalized\,Fitness_{barcode} = \frac{Fitness_{barcode} - Fitness_{DHFR(-)}}{Fitness_{DHFR(+)} - Fitness_{DHFR(-)}},$$

where $Fitness_{barcode}$ is the fitness of each double barcoded PPiSeq strain, $Fitness_{DHFR(-)}$ is the mean fitness of 100 control strains that lack a mDHFR reporter, and $Fitness_{DHFR(+)}$ is the mean fitness of 10 control strains that have a full-length mDHFR reporter under the *TEF1* promoter. We considered all barcodes that mark the same PPI from two datasets as replicate measurements for that PPI. To compare PPIs called in the SD environment with other environments (*Figure 2* and after), we removed 783 PPIs that were called using a single barcode (measured twice in SD, but only once in all other environments).

### Calculating a p-value for each protein-protein pair

To identify positive PPIs in each environment, we first generated a null fitness distribution from control strains (ORF x Null; 17,594 double barcodes) that contain a DHFR-fragment control (F[1,2] or F[3]) that is not fused with an ORF paired with an ORF-DHFR fusion (F[3] or F[1,2], respectively). For each protein pair with at least 2 fitness scores (2 double barcodes), we compared the replicate fitness scores against this null distribution using a one-sided Welch's t-test and obtained a p-value for each protein pair. Protein pairs with only one fitness score were not considered for PPI detection

### Construction of positive reference sets

A positive reference set in each environment was constructed by identifying high-confidence PPIs from the BioGRID database (BIOGRID-organism-Saccharomyces_cerevisiae_S288C-3.4.160) (*Stark et al., 2006*). We defined any protein pair identified as a PPI by at least three separate methods, and at least one binary method ('Two-hybrid', 'FRET', 'PCA'), to be a high-confidence PPI. For each environment, we selected this high-confidence set from all protein pairs with at least two fitness scores in that environment (i.e. it has a p-value determined in section of calculating a p-value for each protein-protein pair) as the positive reference set (SD1: 580 protein pairs; SD2: 586; SD-merge: 633; Forskolin: 534; FK506: 517; NaCl: 563; Raffinose: 432; Hydroxyurea: 576; $H_2O_2$: 585; Doxorubicin: 561; 16℃: 578). We note that most previously defined PPIs have been observed in standard growth media (e.g. SD), so we expect that some fraction of positive reference PPIs may not be present in other environments. In the absence of any previous environment-specific PPI calls, these positive reference sets were nevertheless useful for selecting thresholds for our PPI calling (see below).

### Construction of random reference sets

Fifty random reference sets were constructed in each environment by sampling from all protein pairs that we assayed with no evidence of being a PPI in the BioGRID database. Because most protein

protein pairs are not expected to physically interact, we used this random reference set as putative negative PPIs for selecting thresholds for our PPI calling. The number of protein pairs selected for the random reference set was chosen to reflect the percentage of protein pairs in our screen that have been identified as a PPI in BioGRID by any form of evidence (~1%). That is, in an environment with 600 protein pairs in its positive reference set, we selected ~60,000 (100x) protein pairs for each random reference set. Choosing a random reference set of this size allows us to more accurately report positive predictive values (*Jensen and Bork, 2008*).

We note that the number of PPIs included in the positive and negative reference sets impact positive predictive values (PPVs), and thus PPVs between studies that use different sized reference sets are not directly comparable (*Jensen and Bork, 2008*). For example, the ratio of number of PPIs in the positive reference set over the negative reference set is ~40 times smaller in the proteome-scale PCA study, resulting in higher PPVs (*Jensen and Bork, 2008*; *Tarassov et al., 2008*). If the proteome-scale PCA study used similar sized reference sets as we do here, the PPV for positive calls would be comparable to this study.

## Calling positive PPIs using a dynamic threshold on mean fitness and p-value

PPIs were identified in each environment using a combination of the mean fitness (*f*) and p-value ($Log_{10}$(p-value): *p*) for each protein pair in that environment. To determine dynamic *f* and *p* thresholds for a PPI call, we first chose many discrete threshold combinations of *f* (0 to 0.5) and *p* (−5 to 0) and determined the PPV for each using the positive and random reference sets to estimate true and false positives:

$$PPV = \frac{TP}{TP + FP}$$

where TP is the number of true positives. FP is the number of false positives.

Many combinations of discrete *f* and *p* values have a similar PPV (*Appendix 1—figure 1A*). However, the PPV for a given *f* and *p* can vary when using different random reference sets (*Appendix 1—figures 1B* and *2*). Therefore, to determine the mean relationship between *f* and *p* for each PPV in an environment, we fit a sigmoidal curve [nls(*f* ~ SSlogis(*p*, Asym, xmid, scal)) in R] using data from all 50 random reference sets in that environment. Data at $p < −4$ (p-value<$10^{-4}$) was generally too sparse to confidently fit a regression. We therefore applied a more conservative constant *f* threshold at these *p* (threshold equals the minimum *f* when *p* >= −4). We next considered each PPV regression line as a potential dynamic threshold for PPI calling. To determine the optimal dynamic threshold for each environment (the PPV regression that renders the best balance of precision and recall), we calculated the number of true positives (TP), true negatives (TN), false positives (FP), and false negatives (FN). The threshold with the maximal Matthews correlation coefficient (MCC) (*Matthews, 1975*) was chosen.

$$MCC = \frac{TP \times TN - FP \times FN}{\sqrt{(TP + FP)\,(TP + FN)\,(TN + FP)\,(TN + FN)}}$$

The chosen threshold and its performance in each environment are shown in *Appendix 1—figure 2* and *Appendix 1—table 1*. The precision and recall plots are shown in *Appendix 1—figure 3*. In all cases, using a dynamic threshold showed superior performance over any combination of discrete thresholds (*Appendix 1—figure 4*).

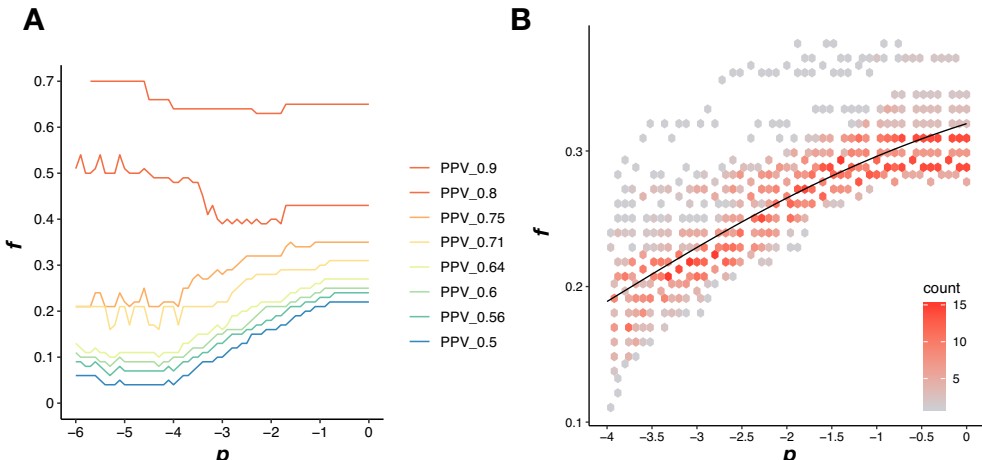

**Appendix 1—figure 1.** Defining a dynamic threshold for PPI calling. (**A**) A discrete combination of a fitness threshold (f) and a p-value threshold (p) results in a PPV. Colored lines are fitness and p-value thresholds that result in the same PPV in SD. (**B**) Density plot of all *f* and *p* combinations that result in a PPV of 0.7 using 50 different random reference sets in SD. The black line is the fitted sigmoid model that is used for the dynamic threshold.

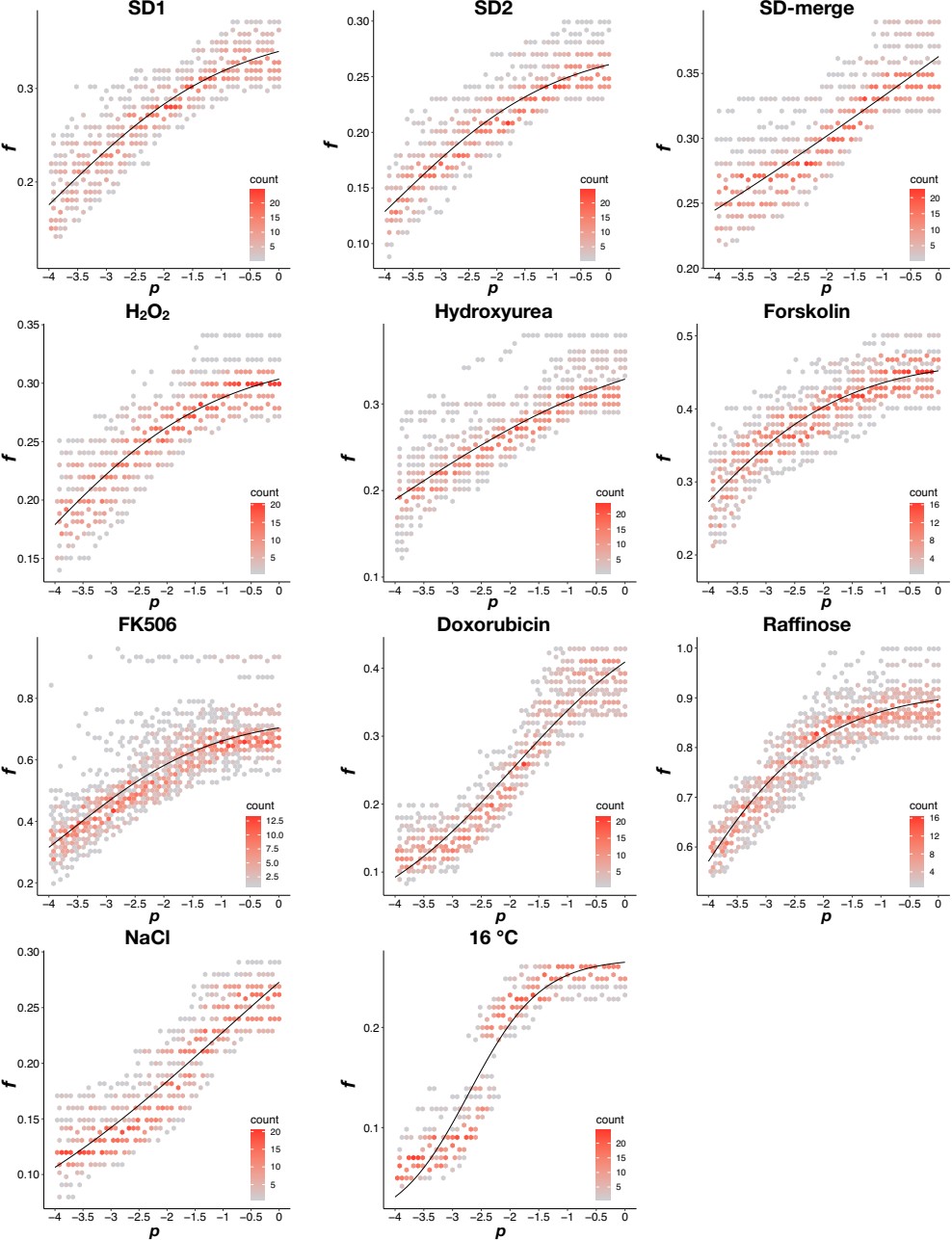

**Appendix 1—figure 2.** Density plots of the dynamic thresholds in each environment. Data were split into two groups: $p < -4$ and $p >= -4$. For $p >= -4$, as in *Appendix 1—figure 1B, a* sigmoidal function was fit to $f$ and $p$ combinations that result in the same PPV value. For $p < -4$, the fitness threshold was set to equal the minimum fitness value when $p >= -4$. The dynamic threshold that results in the maximum MCC in each environment was shown in the plot.

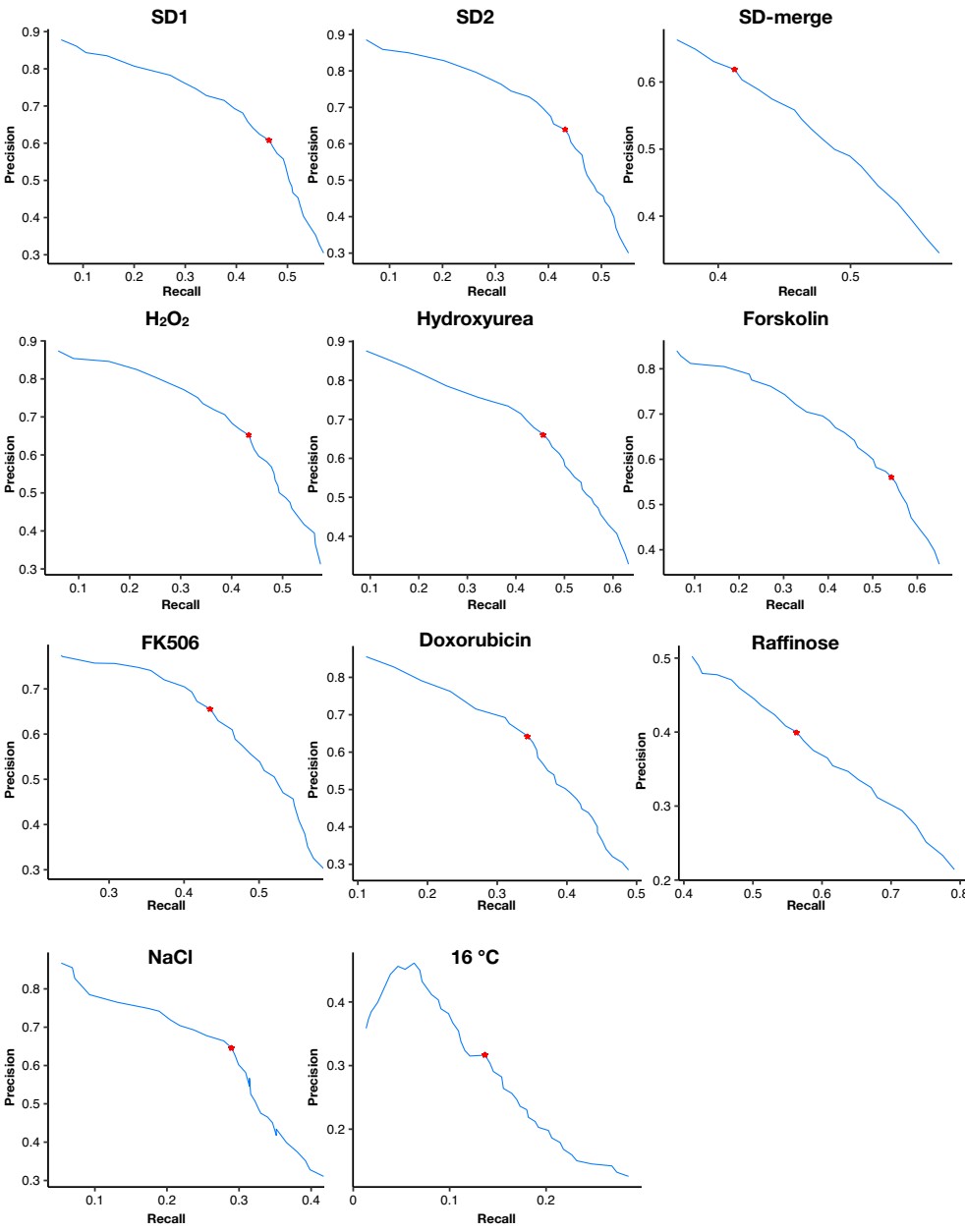

**Appendix 1—figure 3.** The precision-recall curves of dynamic thresholds in each environment. Red asterisks mark the thresholds with maximal Matthews correlation coefficients.

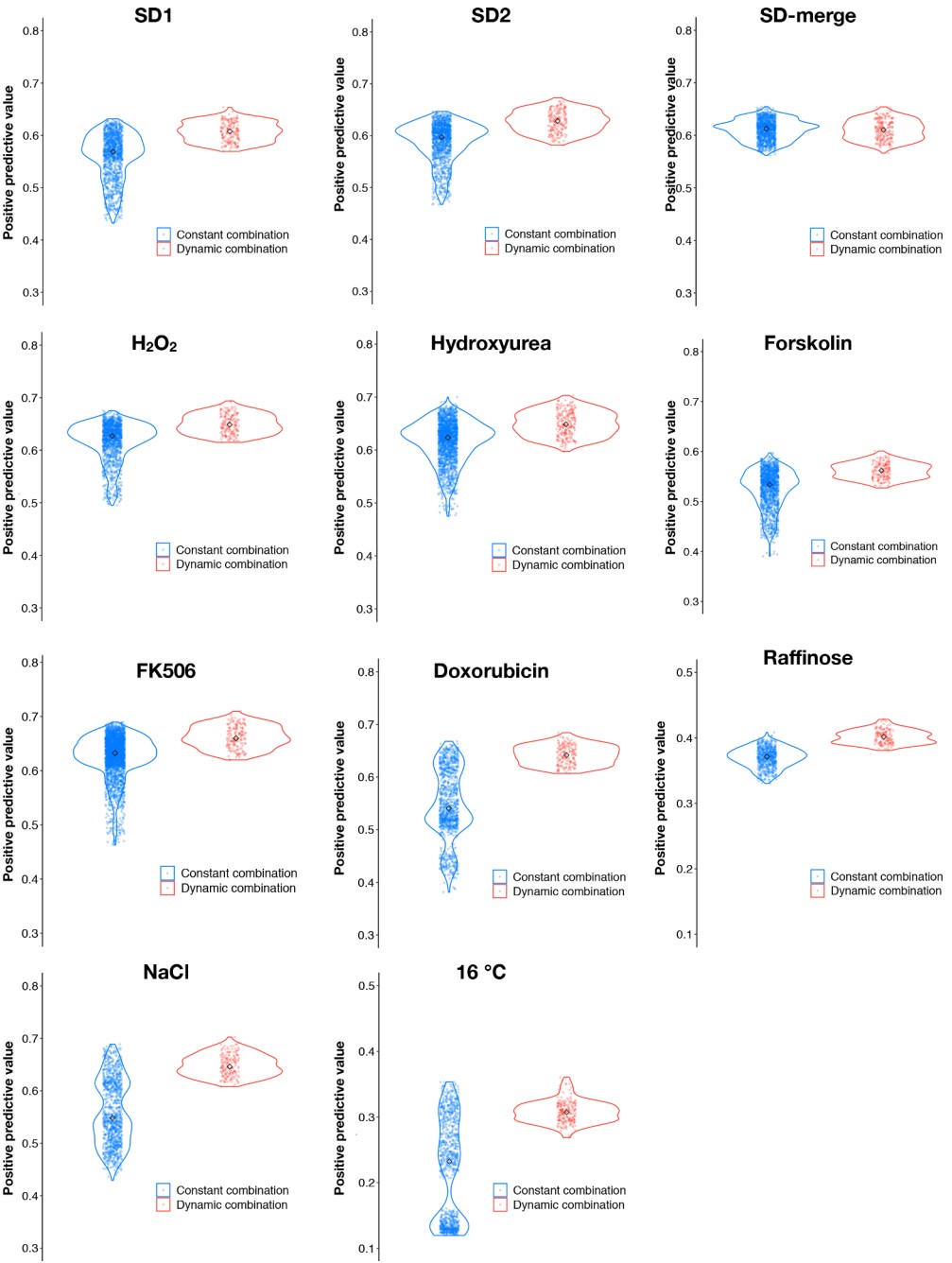

**Appendix 1—figure 4.** Dynamic thresholds (red) of $f$ and $p$ have a higher positive predictive value (PPV) than most discrete combinations (blue). Points represent the PPVs for dynamic thresholding and for all combinations of discrete fitness and p-value thresholds underlying a constant range of false positive rates obtained from the optimal dynamic threshold in each environment.

To examine whether our analyses were robust to different thresholds, we also defined a conservative dynamic threshold with an FPR <0.1%. This strict threshold identifies 'higher confidence' PPIs in each environment (*Appendix 1—table 1*).

**Appendix 1—table 1.** Metrics for the dynamic thresholds used in each environment.
'FPR': false positive rate; 'TPR': true positive rate; 'PPV': positive predictive value; 'MCC': Matthews correlation coefficient; 'Detected_PRS(70)': 70 likely protein interaction pairs in a positive reference set; 'Detected_RRS(67)': 67 random pairs in a random reference set (*Liu et al., 2019*; *Yu et al., 2008*).

**Optimal dynamic threshold based on the best balance between precision and recall**

| Environment | Optimal_threshold | FPR | TPR | PPV | MCC | F1_Score | Detected_PRS (70) | Detected_RRS (67) |
|---|---|---|---|---|---|---|---|---|
| SD1 | 0.7 | 0.00283 | 0.4647 | 0.6075 | 0.5274 | 0.5266 | 20 | 3 |
| SD2 | 0.73 | 0.002315 | 0.4317 | 0.6386 | 0.5214 | 0.5152 | 20 | 2 |
| SD-merge | 0.7 | 0.002473 | 0.4124 | 0.6187 | 0.5012 | 0.4949 | 19 | 3 |
| FK506 | 0.72 | 0.00196 | 0.4352 | 0.6551 | 0.5307 | 0.523 | 20 | 3 |
| H2O2 | 0.73 | 0.002232 | 0.4342 | 0.6517 | 0.5283 | 0.5212 | 20 | 3 |
| Hydroxyurea | 0.74 | 0.00222 | 0.4569 | 0.6613 | 0.5461 | 0.5405 | 19 | 1 |
| NaCl | 0.73 | 0.00145 | 0.2895 | 0.6462 | 0.4292 | 0.3999 | 18 | 1 |
| Forskolin | 0.64 | 0.003727 | 0.5424 | 0.5608 | 0.5476 | 0.5514 | 22 | 2 |
| Raffinose | 0.48 | 0.008105 | 0.5633 | 0.4001 | 0.4688 | 0.4679 | 20 | 2 |
| Doxorubicin | 0.77 | 0.001765 | 0.344 | 0.6425 | 0.4667 | 0.4481 | 18 | 2 |
| 16 ℃ | 0.41 | 0.00281 | 0.1367 | 0.3164 | 0.203 | 0.1909 | 21 | 3 |

**Arbitrary strict dynamic threshold in each environment**

| Environment | Optimal_threshold | FPR | TPR | PPV | MCC | F1_Score | Detected_PRS (70) | Detected_RRS (67) |
|---|---|---|---|---|---|---|---|---|
| SD-merge | 0.79 | 0.000945201 | 0.283949447 | 0.745351747 | 0.457089253 | 0.411234763 | 17 | 2 |
| FK506 | 0.78 | 0.000978346 | 0.338491296 | 0.747573979 | 0.500365002 | 0.465989091 | 18 | 2 |
| H2O2 | 0.8 | 0.000873036 | 0.306051282 | 0.771675172 | 0.48312128 | 0.438278516 | 17 | 2 |
| Hydroxyurea | 0.8 | 0.000984872 | 0.322118056 | 0.75660523 | 0.490767592 | 0.45186047 | 17 | 1 |
| NaCl | 0.76 | 0.000965072 | 0.237371226 | 0.692789678 | 0.402579452 | 0.353591157 | 18 | 1 |
| Forskolin | 0.77 | 0.000921497 | 0.302696629 | 0.742709167 | 0.471420905 | 0.430101999 | 17 | 1 |
| Raffinose | 0.56 | 0.004453894 | 0.426481481 | 0.479253019 | 0.447039993 | 0.451329915 | 18 | 1 |
| Doxorubicin | 0.82 | 0.000989351 | 0.26916221 | 0.71558061 | 0.435929061 | 0.391182865 | 17 | 2 |
| 16 ℃ | 0.51 | 0.000900186 | 0.071384083 | 0.432281211 | 0.172518262 | 0.122533747 | 17 | 3 |

## Detection and removal of promiscuous proteins

We excluded from our PPiSeq library any proteins that have been previously found to promiscuously form (likely spurious) PPIs in standard growth conditions (*Tarassov et al., 2008*). However, new promiscuous proteins may arise in other conditions screened here. We detected these promiscuous bait or prey constructs by determining if a PPI was called when it was paired with a DHFR-fragment control . If a bait or prey construct was identified as promiscuous in one environment, all PPIs containing that construct were removed in that environment. If a bait or prey construct was identified as promiscuous in two or more environments, all PPIs containing that construct were removed from all environments. Promiscuous proteins are summarized in *Appendix 1—table 2*.

**Appendix 1—table 2.** Summary of promiscuous proteins that interact with an mDHFR fragment that is not tethered to any protein.

Promiscuous and non-promiscuous proteins are represented by 1 and 0, respectively, in each environment.

| PPI | Positive_environme_number | SD_merge | $H_2O_2$ | Hydroxyurea | Doxorubicin | Forskolin | Raffinose | NaCl | FK506 | 16 ℃ |
|---|---|---|---|---|---|---|---|---|---|---|
| YMR120C | 6 | 1 | 1 | 1 | 0 | 1 | 1 | 0 | 1 | 0 |
| YIL143C | 6 | 1 | 1 | 1 | 0 | 1 | 1 | 1 | 0 | 0 |
| YLL034C | 5 | 1 | 1 | 0 | 1 | 0 | 0 | 1 | 0 | 1 |
| YPL139C | 4 | 1 | 1 | 0 | 0 | 0 | 0 | 1 | 1 | 0 |

*Continued on next page*

*Appendix 1—table 2 continued*

| PPI | Positive_environme_number | SD_merge | H$_2$O$_2$ | Hydroxyurea | Doxorubicin | Forskolin | Raffinose | NaCl | FK506 | 16 °C |
|---|---|---|---|---|---|---|---|---|---|---|
| YGR278W | 2 | 0 | 1 | | 0 | 0 | 0 | 1 | 0 | 0 |
| YPL112C | 2 | 0 | 0 | 0 | 1 | 0 | 0 | 1 | 0 | 0 |
| YIL070C | 2 | 0 | 0 | 0 | 1 | 0 | 0 | 0 | 1 | 0 |
| YHL007C | 2 | 0 | 0 | 0 | 1 | 0 | 0 | 0 | 1 | 0 |
| YDR452W | 2 | 0 | 0 | 0 | 1 | 0 | 0 | 0 | 1 | 0 |
| YOL147C | 1 | 0 | 1 | 0 | 0 | 0 | 0 | 0 | 0 | 0 |
| YER087W | 1 | 0 | 0 | 0 | 1 | 0 | 0 | 0 | 0 | 0 |
| YER063W | 1 | 0 | 0 | 0 | 1 | 0 | 0 | 0 | 0 | 0 |
| YOR323C | 1 | 0 | 0 | 0 | 1 | 0 | 0 | 0 | 0 | 0 |
| YNL064C | 1 | 0 | 0 | 0 | 1 | 0 | 0 | 0 | 0 | 0 |
| YKR080W | 1 | 0 | 0 | 0 | 0 | 1 | 0 | 0 | 0 | 0 |
| YJL153C | 1 | 0 | 0 | 0 | 0 | 1 | 0 | 0 | 0 | 0 |
| YDL208W | 1 | 0 | 0 | 0 | 0 | 0 | 0 | 1 | 0 | 0 |
| YLR182W | 1 | 0 | 0 | 0 | 0 | 0 | 0 | 1 | 0 | 0 |
| YPR124W | 1 | 0 | 0 | 0 | 0 | 0 | 0 | 1 | 0 | 0 |
| YHR114W | 1 | 0 | 0 | 0 | 0 | 0 | 0 | 1 | 0 | 0 |
| YDR381C-A | 1 | 0 | 0 | 0 | 0 | 0 | 0 | 1 | 0 | 0 |
| YGR198W | 1 | 0 | 0 | 0 | 0 | 0 | 0 | 1 | 0 | 0 |
| YDR171W | 1 | 0 | 0 | 0 | 0 | 0 | 0 | 1 | 0 | 0 |
| YGR130C | 1 | 0 | 0 | 0 | 0 | 0 | 0 | 1 | 0 | 0 |
| YLL022C | 1 | 0 | 0 | 0 | 0 | 0 | 0 | 1 | 0 | 0 |
| YMR136W | 1 | 0 | 0 | 0 | 0 | 0 | 0 | 1 | 0 | 0 |
| YKL010C | 1 | 0 | 0 | 0 | 0 | 0 | 0 | 1 | 0 | 0 |
| YCR033W | 1 | 0 | 0 | 0 | 0 | 0 | 0 | 1 | 0 | 0 |
| YPL083C | 1 | 0 | 0 | 0 | 0 | 0 | 0 | 1 | 0 | 0 |
| YOR360C | 1 | 0 | 0 | 0 | 0 | 0 | 0 | 1 | 0 | 0 |
| YOR393W | 1 | 0 | 0 | 0 | 0 | 0 | 0 | 1 | 0 | 0 |
| YNL026W | 1 | 0 | 0 | 0 | 0 | 0 | 0 | 1 | 0 | 0 |
| YGR195W | 1 | 0 | 0 | 0 | 0 | 0 | 0 | 1 | 0 | 0 |
| YOR306C | 1 | 0 | 0 | 0 | 0 | 0 | 0 | 1 | 0 | 0 |
| YDL093W | 1 | 0 | 0 | 0 | 0 | 0 | 0 | 1 | 0 | 0 |
| YCR059C | 1 | 0 | 0 | 0 | 0 | 0 | 0 | 1 | 0 | 0 |
| YOL081W | 1 | 0 | 0 | 0 | 0 | 0 | 0 | 1 | 0 | 0 |
| YGR140W | 1 | 0 | 0 | 0 | 0 | 0 | 0 | 1 | 0 | 0 |
| YKL139W | 1 | 0 | 0 | 0 | 0 | 0 | 0 | 1 | 0 | 0 |
| YEL017C-A | 1 | 0 | 0 | 0 | 0 | 0 | 0 | 0 | 1 | 0 |
| YDL112W | 1 | 0 | 0 | 0 | 0 | 0 | 0 | 0 | 1 | 0 |
| YDR057W | 1 | 0 | 0 | 0 | 0 | 0 | 0 | 0 | 1 | 0 |
| YFR001W | 1 | 0 | 0 | 0 | 0 | 0 | 0 | 0 | 0 | 1 |
| YJL124C | 1 | 0 | 0 | 0 | 0 | 0 | 0 | 0 | 0 | 1 |
| YDR151C | 1 | 0 | 0 | 0 | 0 | 0 | 0 | 0 | 0 | 1 |
| YBR057C | 1 | 0 | 0 | 0 | 0 | 0 | 0 | 0 | 0 | 1 |

*Continued on next page*

*Appendix 1—table 2 continued*

| PPI | Positive_environme_ number | SD_merge | H$_2$O$_2$ | Hydroxyurea | Doxorubicin | Forskolin | Raffinose | NaCl | FK506 | 16 ℃ |
|-----|------------|----------|------|-------------|-------------|-----------|-----------|------|-------|------|
| YHR146W | 1 | 0 | 0 | 0 | 0 | 0 | 0 | 0 | 0 | 1 |
| YDR379W | 1 | 0 | 0 | 0 | 0 | 0 | 0 | 0 | 0 | 1 |
| YDR513W | 1 | 0 | 0 | 0 | 0 | 0 | 0 | 0 | 0 | 1 |
| YMR227C | 1 | 0 | 0 | 0 | 0 | 0 | 0 | 0 | 0 | 1 |
| YDR420W | 1 | 0 | 0 | 0 | 0 | 0 | 0 | 0 | 0 | 1 |

# Appendix 2

## Supplementary tables

Tables can be downloaded from: https://osf.io/jmhrb/ (*Liu, 2020b*).

Table S1. Bottleneck cell size and generation number of serial batch culture in different environments. 'T-1' is frozen stock from which all competitions start. 'T0' to 'T7' are the serial population bottlenecks.

Table S2. Read count of each double barcode at different time points in each environment. Four or five time points were sequenced per environment. Spike-in control PPIs were named as follows in the column 'PPI': 'ORF_HO:TEF1pr-DHFR1-2', 'ORF_HO:TEF1pr-linker-DHFR1-2', 'HO:TEF1pr-DHFR3_ORF', and 'HO:TEF1pr-linker-DHFR3_ORF' are interactions with a DHFR fragment that is not tethered to a yeast protein (ORF X Null in *Figure 1D*); 'positive_DHFR' are yeast strains that contain a full length mDHFR under a strong promoter (DHFR + in *Figure 1D*); 'negative_non_DHFR' are yeast strains that lack any mDHFR fragment (DHFR- in *Figure 1D*); 'Pos_PPI_number-first (ORF1 ~Pos_ORF2 ~Pos)' and 'Pos_PPI_number-second(ORF2 ~Pos_ORF1 ~Pos)' are 70 likely protein interaction pairs; 'Neg_PPI_number-first(ORF1 ~Neg_ORF2 ~Neg)' and 'Neg_PPI_number-second(ORF2 ~Neg_ORF1 ~Neg)' are 67 random pairs. These likely protein interaction pairs and random pairs were chosen from the previously constructed reference sets (*Liu et al., 2019*; *Yu et al., 2008*). The suffixes of 'first' and 'second' stand for the same protein pair with a different protein chimera acting the bait protein (i.e. ORF1-DHFR[1,2] X ORF2-DHFR[3] and ORF1-DHFR[3] X ORF2-DHFR[1,2]).

Table S3. Estimated fitness and estimation error of each double barcode in each environment. Spike-in controls are as in Appendix 2 Table S2. The estimation error, *d*, describes the deviation of the predicted counts from the observed counts at different time points for each double barcode (see Materials and methods). High *d* indicates higher error in fitness estimation.

Table S4. Number of double barcodes found in the pool, mean fitness, p-value, and whether or not a PPI is called. For PPI calling, one represents a positive PPI and 0 represents a negative PPI. Spike-in controls are as in Appendix 2 Table S2.

Table S5. Predicted validation rates for identified PPIs in each environment.

Table S6. High-confidence PPIs in each environment. In each environment, one represents a positive PPI and 0 represents a negative PPI.

Table S7. Significant gene ontology terms enriched for proteins in each community of the PPI network as detected by the InfoMAP algorithm.

Table S8. Protein features derived from the PPiSeq multi-environment network and other gene features. 'PPI.degree.PPiSeq' is the number of PPIs detected for each protein. 'Mean.positive.environment.number.PPiSeq' is the mean number of environments in which those PPIs are detected. 'Standard.deviation.positive.environment.number.PPiSeq' is the standard deviation of the number of environments in which those PPIs are detected.

