## [Decision Letter]

[Editors' note: this paper was reviewed by Review Commons.]

**Acceptance summary:**

The organization of protein-protein interaction networks has mainly been described in standard laboratory conditions. Because of this, we do not know how static and how complete these networks are. Here, the authors combine an in vivo protein-protein interaction detection assay with barcode sequencing to interrogate more than one million potential interactions across multiple environments. They show that some interactions are stable while others are condition-dependent, and they identify protein properties associated with each category. Importantly, they show that our current knowledge of these networks cannot be complete unless we study their organization in multiple conditions.

---

## [Author Response]

Reviewer #1 (Evidence, reproducibility and clarity):The manuscript is clearly written and the figures appropriate and informative. Some descriptions of data analyses are a little dense but reflect what would appear long hard efforts on the part of the authors to identify and control for possible sources of misinterpretation due to sensitivities of parameters in their fitness model. The authors efforts to retest interactions under non-competition conditions allay fears of most concerns that I would have. One problem though that I could not see explicitly addressed was that of potential effects of interactions between methotrexate and the other conditions and how this is controlled for. Specifically, I could be argued that the fact that a particular PPI is observed under a specific condition could have more to do with a synthetic effect of treatment of cells with a drug plus methotrexate. Is this controlled for and how? I raise this because in a chemical genetic screen for fitness it was shown that methotrexate is particularly promiscuous for drug-drug interactions (Hillenmeyer ME et al., Science 2008). I tried to think of how this works but couldn't come up with anything immediately. I'd appreciate if the authors would take a crack at resolving this issue. Otherwise I have no further concerns about the manuscript.

We thank the reviewer for the kind comments. We agree with the reviewer’s point that methotrexate could be interacting with drugs or other perturbagens, similar to how the chosen nitrogen source, carbon source, or other growth conditions may interact with a drug. However, the methotrexate concentration is held constant across all conditions, as is the rest of the media components such as the nitrogen and carbon source (with the exception of the raffinose perturbation). Any interactions with methotrexate, or other media components, is undetectable without systematically varying all components for all stressors. Therefore, we use the typical experimental design of measuring molecular variation from a reference, holding invariant media components (such as methotrexate, glucose, or vitamins) fixed between conditions. This is a general practice, and we describe that every condition contains methotrexate:

“The library was grown under mild methotrexate selection in 9 environments for 12-18 generations in serial batch culture, diluting 1:8 every ~3 generations, with a bottleneck population size greater than 2 x 10^9^ cells (Table S1).”

We also list the full details of each environment in Table S1.

Reviewer #1 (Significance):Lui et al. expand on previous work from the Levy group to explore a massive in vivo protein interactome in the yeast *S. cerevisiae*. They achieve this by performing screens cross 9 growth conditions, which, with replication, results in a total of 44 million measurements. Interpreting their results based on a fitness model for pooled growth under methotrexate selection, they make the key observation that there is a vastly expanded pool of protein-protein interactions (PPI) that are found under only one or two condition compared to a more limited set of PPI that are found under a broad set of conditions (mutable versus immutable interactors). The authors show that this dichotomy suggests some important features of proteins and their PPIs that raise important questions about functionality and evolution of PPIs. Among these are that mutable PPIs are enriched for cross-compartmental, high disorder and higher rates of evolution and subcellular localization of proteins to chromatin, suggesting roles in gene regulation that are associated with cellular responses to new conditions. At the same time these interactions are not enriched for changes in abundance. These results are in contrast to those of immutable PPIs, which seem to form a core background noise, more determined by changes in abundance than what the authors interpret must be post-translational processes that may drive, for instance, changes in subcellular localization resulting in appearance of PPIs under specific conditions. The authors are also able to address a couple of key issues about protein interactomes, including the controversial Party-date Hub hypothesis of Vidal, in which they could now affirm support for this hypothesis based on their results and notably negative correlation of PPIs to protein abundance for mutable PPIs. Finally, they also addressed the problem of predicting the upper limit of PPIs in yeast, showing the remarkable results that it may be no more than about 2 times the number of proteins expressed by yeast. Such an upper limit is profoundly important to modelling cellular network complexity and, if it holds up, could define a general upper limit on organismal complexity.This manuscript is a very important contribution to understanding dynamics of molecular networks in living cells and should be published with high priority.Reviewer #2 (Evidence, reproducibility and clarity):Report on Liu et al. "A large accessory protein interactome is rewired across environments" Liu et al. use a mDHFR-based, pooled barcode sequencing / competitive growth / mild methotrexate selection method to investigate changes of PPI abundance of 1.6 million protein pairs across different 9 growth conditions. Because most PPI screens aim to identify novel PPIs in standard growth conditions, the currently known yeast PPI network may be incomplete. The key concept is to define immutable PPIs that are found in all conditions and "mutable" PPIs that are present in only some conditions.The assay identified 13764 PPIs across the 9 conditions, using optimized fitness cut offs. Steady PPI i.e. across all environments, were identified in membrane compartments and cell division. Processes associated with the chromosome, transcription, protein translation, RNA processing and ribosome regulation were found to change between conditions. Mutable PPIs are form modules as topological analyses reveals.Interestingly, a correlation on intrinsic disorder and PPI mutability was found and postulated as more flexible in the conformational context, while at the same time they are formed by less abundant proteins.I appreciate the trick to use homodimerization as an abundance proxy to predict interaction between heterodimers (of proteins that homodimerize). This "mass-action kinetics model" explains the strength of 230 out of 1212 tested heterodimers.A validation experiment of the glucose transporter network was performed and 90 "randomly chosen" PPIs that were present in the SD environment were tested in NaCl (osmotic stress) and Raffinose (low glucose) conditions through recording optical density growth trajectories. Hxt5 PPIs stayed similar in the tested conditions, supported by the current knowledge that Hxt5 is highly expressed in stationary phase and under salt stress. In Raffinose, Hxt7, previously reported to increase the mRNA expression, lost most PPIs indicating that other factors might influence Hxt7 PPIs.Points for consideration:A clear definition of mutable and immutable is missing or could not be found.

We thank the reviewer for pointing this out. We have now added better definition of mutable and immutable in subsection “A large dynamic accessory protein interactome”:

“We partitioned PPIs by the number of environments in which they were identified and defined PPIs at opposite ends of this spectrum as “mutable” PPIs (identified in only 1-3 environments) and “immutable” (identified in 8-9 environments).”

Approximately half of the PPIs have been identified in one environment. Many of those mutable PPIs were detected in the 16°C condition. Is there an explanation for the predominance of this specific environment? What are these PPIs about?

The reviewer is correct that ~40% of the PPIs identified in only one environment were found in the 16℃ environment. One reason for this could be technical: the positive predictive value (PPV) is the lowest amongst the conditions (16℃: 31.6%, mean: 57%, Appendix 1—table 1). It must be noted, however, that PPVs are calculated using reference data that has generally been collected in standard growth conditions. So, it might be expected that the most divergent environment from standard growth conditions (resulting in the most differences in PPIs) would result in a lower PPV in our study even if the true frequency of false positives was equivalent across environments. We have attempted to be transparent about the quality of the data in each environment by reporting PPVs and other metrics in Appendix 1—table 1. However, we suspect that the large number of PPIs unique to 16°C is due in part to the fact that it causes the largest changes in the protein interactome, and believe that it should be included, even at the risk of lowering the overall quality of the data. The main reason for this is that this data is likely to contain valuable information about how the cell copes with this stress. For example, we find, but do not highlight in the manuscript, that 16°C-specific PPIs contain two major hubs (DID4: 285 PPIs involved in endocytosis and vacuolar trafficking, and DED1: 102 PPIs involved in translation), both of which are reported to be associated with cold adaptation in yeast (Hilliker et al., 2011; Isasa et al., 2015).

To assess whether the potentially higher false-positive rate in 16°C could be impacting our conclusions related to PPI network organization and features of immutable and mutable PPIs, we repeated these analyses leaving out the 16°C data and found that our main conclusions did not change. This new analysis is now presented in Figure 3—figure supplement 6 and described in subsection “A large dynamic accessory protein interactome”.

“Finally, we used a pair of more conservative PPI calling procedures that either identified PPIs with a low rate of false positives across all environments (FPR < 0.1%) or removed data from the most extreme environmental perturbation (16°C, see Materials and methods, Table S6). Using these higher confidence sets, we still found that mutable PPIs far outnumbered others in the multi-condition PPI network (Figure 3—figure supplement 4).”

We have also added references to other panels in Figure 3—figure supplement 4 throughout the manuscript, where appropriate.

50 % overall retest validation rate is fair and reflects a value comparable to other large-scale approaches. However, what is the actual variation, e.g. between mutable PPIs and immutable or between condition. e.g. at 16°C.

We validated 502 PPIs present in the SD environment and an additional 36 PPIs in the NaCl environment. As the reviewer suggests, we do indeed observe differences in the validation rate across mutability bins. This data is reported in Figures 3B and Figure 3—figure supplement 4B, and we use this information to provide a confidence score for each PPI in subsection “A large dynamic accessory protein interactome”.

“To better estimate how the number of PPIs changes with PPI mutability, we used these optical density assays to model the validation rate as a function of the mean PPiSeq fitness and the number of environments in which a PPI is detected. […] Using this more conservative estimate, we still found a preponderance of mutable PPIs (Figure 3—figure supplement 4).”

The validation rate in NaCl is similar to SD (39%, 14/36), suggesting that validation rates do not vary excessively across environments. Because validation experiments are time consuming (we performed 6 growth experiments per PPI), performing a similar scale of validations in all environments as in SD would be resource intensive. Instead, we report a number of metrics (true positive rate, false positive rate, positive predictive value) in Appendix 1—table 1 using large positive and random reference sets. We believe these metrics are sufficient for readers to compare the quality of data across environments.

What is the R correlation cut-off for PPIs explained in the mass equilibrium model vs. not explained?

We do not use an R correlation cut-off to assess if a PPI is explained by the mass-action equilibrium model. We instead rely on ordinary least-squares regression as detailed in the Materials and methods.

“…we used ordinary least-squares linear regression in R to fit a model of the geometric mean of the homodimer signals multiplied by a free constant and plus a free intercept.

[…] This criteria was used to identify PPIs for which protein expression does or does not appear to play as significant of a role as other post-translational mechanisms.”

The first criterion identifies a quantitative fit to the model of variation being related. The second criterion is used to filter out PPIs for which the relationship appears to be explained by more than just the homodimer signals. This approach is more stringent, but we believe this is the most appropriate statistical test to assess fit to this linear model.

90 "randomly chosen" PPIs for validation. It needs to be demonstrated that these interactions are a random subset otherwise is could also mean cherry picked interactions.

We selected 90 of the 284 glucose transport-related PPIs for validation using the “sample” function in R (replace = FALSE). We have now included text that describes this in the Materials and methods:

“Diploids (PPIs) on each plate were randomly picked using the “sample” function in R (replace = FALSE) from PPIs that meet specific requirements.”

Figure 4 provides interesting correlations with the goal to reveal properties of mutable and less mutable PPIs. PPIs detected in the PPIseq screen can partially be correlated to co-expression (4A) as well as co-localization. Does it make sense to correlate the co-expression across number of conditions? Are the expression correlation conditions specific? In this graph it could be that expression correlation stems from condition 1 and 2 and the interaction takes place in 4 and 5 still leading to the same conclusion.… Is the picture of the co-expression correlation similar when you simply look at individual environments like in S4A?

We use co-expression mutual rank scores from the COXPRESdb v7.3 database (Obayashi et al., 2019). These mutual rank scores are derived from a broad set of 3593 environmental perturbations that are not limited to the environments we tested here. By using this data, we are asking if co-expression in general is correlated with mutability and report that it is in Figure 4A. We thank the reviewer for pointing out that this was not clear and have now added text to clarify that the co-expression analysis is derived from external data:

“We first asked whether co-expression is indeed a predictor of PPI mutability and found that it is: co-expression mutual rank (which is inversely proportional to co-expression across thousands of microarray experiments) declined with PPI mutability (Figures 4A and Figure 4—figure supplement 1) (Obayashi and Kinoshita, 2009; Obayashi et al., 2019).”

The new Figure 4—figure supplement 1 examines how the co-expression mutual rank changes with PPI mutability for PPIs identified in each environment, as the reviewer suggested. For each environment, we find the same general pattern as in Figure 4A (which considers PPIs from all environments).

Figure 4C: Interesting, how dependent are the various categories?

It is well known that many of these categories are correlated (e.g. mRNA expression level and protein abundance, and deletion fitness effect and genetic interaction degree). However, we believe it is most valuable to report the correlation of each category with PPI mutability independently in Figures 4C and Figure 4—figure supplement 4, since similar correlations with related categories provide more confidence in our conclusions.

Figure 4 F: When binned in the number of environments in which the PPI was found, the distribution peaks at 6 environments and decreases with higher and lower number of environments. The description /explanation in the text clearly says something else.

We reported in subsection:

“We next used logistic regression to determine what features may underlie a good or poor fit to the model (Figure 4—figure supplement 6C) and found that PPI mutability was the best predictor, with more mutable PPIs being less frequently explained (Figure 4F). Unexpectedly, mean protein abundance was the second-best predictor, with high abundance predicting a poor fit to the model, particularly for less mutable PPIs (Figure 4—figure supplement 6D and Figure 4—figure supplement 6E).”

As the reviewer notes, Figure 4F shows that the percent of heterodimers explained by the model does appear to decrease for PPIs observed in the most environments. We suspect that the reviewer is correct that something more complicated is going on. One possibility is that extraordinarily stable PPIs (stable in all conditions) would have less quantitative variation in protein or PPI abundance across environments. If this is true, it would be statistically difficult to fit the mass action kinetics model for these PPIs (lower signal relative to noise), thereby resulting in the observed dip.

A second possibility is that multiple correlated factors are associated with contributing positively or negatively to a good fit, and the simplicity of Figure 4F or a Pearson correlation does not capture this interplay. This second possibility is why we used multivariate logistic regression (Figure 4—figure supplement 6C) to dissect the major contributing factors. In the text quote above, we report that high abundance is *anti-correlated* with a good fit to the model (Figure 4—figure supplement 6D, Figure 4—figure supplement 6E). Figure 4C shows that immutable PPIs tend to be formed from highly abundant proteins. One possible explanation is that highly abundant proteins saturate the binding sites of their binding partners, breaking from the assumptions of mass action kinetics model. We have now changed the word “limit” to “saturate” in subsection “Properties of mutable PPIs” to make this concept more explicit.

“Taken together, these data suggest that mutable PPIs are subject to more post-translational regulation across environments and that high basal protein abundance may saturate the binding sites of their partners, limiting the ability of gene expression changes to regulate PPIs.”

A third possibility is that the dip is simply due to noise. Given the complexity of the possible explanations and our uncertainty about which is more likely, we chose to leave this description out of the main text and focus on the major finding: that PPIs detected in more environments are *generally* associated with a better fit to the mass action kinetics model.

Figure 6: I apologize, but for my taste this is not a final Figure 6 for this study. Investigation of different environments increases the PPI network in yeast, yes, yet it is very well known that a saturation is reached after testing of several conditions, different methods and even screening repetition (sampling). It does not represent an important outcome. Move to supplement or remove.

We included Figure 6 to summarize and illustrate the path forward from this study. This is an explicit reference to impactful computational analyses done using earlier generations of data to assess the completeness of single-condition interaction networks (Hart et al., 2006; Sambourg and Thierry-Mieg, 2010). Here, we are extending PPI measurement of millions-scale networks across multiple environments and are using this figure to extend these concepts to multi-condition screens. We agree that the property of saturation in sampling is well known, but it is surprising that we can quantitatively estimate convergence of this expanded condition-specific PPI set using only 9 conditions. Thus, we agree with reviewer 1 that these are “remarkable results” and that the “upper limit is profoundly important to modelling cellular network complexity and, if it holds up, could define a general upper limit on organismal complexity.” We think this is an important advance of the paper, and this figure is useful to stimulate discussion and guide future work.

Reviewer #2 (Significance):Liu et al. increase the current PPI network in yeast and offer a substantial dataset of novel PPIs seen in specific environments only. This resource can be used to further investigate the biological meaning of the PPI changes. The data set is compared to previous DHFR providing some sort of quality benchmarking. Mutable interactions are characterized well. Clearly a next step could be to start some "orthogonal" validation, i.e. beyond yeast growth under methotrexate treatment.

The reviewer makes a great point that we also discuss in the Discussion:

“While we used reconstruction of C-terminal-attached mDHFR fragments as a reporter for

PPI abundance, similar massively parallel assays could be constructed with different PCA reporters or tagging configurations to validate our observations and overcome false negatives that are specific to our reporter. Indeed, the recent development of “swap tag” libraries, where new markers can be inserted C- or N-terminal to most genes (Weill et al., 2018; Yofe et al., 2016), in combination with our iSeq double barcoder collection (Liu et al., 2019), makes extension of our approach eminently feasible.”

Reviewer #3 (Evidence, reproducibility and clarity):Summary:The manuscript "A large accessory protein interactome is rewired across environments" by Liu et al. scales up a previously described method (PPiSeq) to test a matrix of ~1.6 million protein pairs of direct protein-protein interactions in each of 9 different growth environments.While the study found a small fraction of immutable PPIs that are relatively stable across environments, the vast majority were “mutable” across environments. Surprisingly, PPIs detected only in one environment made up more than 60% of the map. In addition to a false positive fraction that can yield apparently mutable interactions, retest experiments demonstrate (not surprisingly) that environment-specificity can sometimes be attributed to false-negatives. The study authors predict that the whole subnetwork within the space tested will contain 11K true interactions.Much of environment-specific rewiring seemed to take place in an “accessory module”, which surrounds the core module made of mostly immutable PPIs. A number of interesting network clustering and functional enrichment analyses are performed to characterize the network overall and “mutable” interactions in particular. The study report other global properties such as expression level, protein abundance and genetic interaction degree that differ between mutable and immutable PPIs. One of the interesting findings was evidence that many environmentally mutable PPI changes are regulated post-translationally. Finally, authors provide a case study about network rewiring related to glucose transport.Major issues:The Results section should more prominently describe the dimensions of the matrix screen, both in terms of the set of protein pairs attempted and the set actually screened (I think this was 1741 x 1113 after filtering?). More importantly, the study should acknowledge in the Introduction that this was not a random sample of protein pairs, but rather focused on pairs for which interaction had been previously observed in the baseline condition. This major bias has a potentially substantial impact on many of the downstream analyses. For example, any gene which was not expressed under the conditions of the original Tarrasov et al. study on which the screening space was based will not have been tested here. Thus, the study has systematically excluded interactions involving proteins with environment-dependent expression, except where they happened to be expressed in the single Tarrasov et al. environment. Heightened connectivity within the “core module” may result from this bias, and if Tarrasov et al. had screened in hydrogen peroxide (H_2_O_2_) instead of SD media, perhaps the network would have exhibited a code module in H_2_O_2_ decorated by less-densely connected accessory modules observed in other environments. The paper should clearly indicate which downstream analyses have special caveats in light of this design bias.

We have now added text the matrix dimensions of our study in the Results:

“To generate a large PPiSeq library, all strains from the protein interactome

(mDHFR-PCA) collection that were found to contain a protein likely to participate in at least one PPI (1742 X 1130 protein pairs), (Tarassov et al., 2008) were barcoded in duplicate using the double barcoder iSeq collection (Liu et al., 2019), and mated together in a single pool (Figure 1A). Double barcode sequencing revealed that the PPiSeq library contained 1.79 million protein pairs and 6.05 million double barcodes (92.3% and 78.1% of theoretical, respectively, 1741 X 1113 protein pairs), with each protein pair represented by an average of 3.4 unique double barcodes (Figure 1—figure supplement 1).”

We agree with the reviewer that our selection of proteins from a previously identified set can introduce bias in our conclusions. Our research question was focused on how PPIs change across environments, and thus we chose to maximize our power to detect PPI changes by selecting a set of protein pairs that are enriched for PPIs. We have now added a discussion of the potential caveats of this choice to the Discussion:

“Results presented here and elsewhere (Huttlin et al., 2020) suggest that PPIs discovered under a single condition or cell type are a small subset of the full protein interactome emergent from a genome. […] Nevertheless, results presented here provide a new mechanistic view of how the cell changes in response to environmental challenges, building on the previous work that describes coordinated responses in the transcriptome (Brauer et al., 2007; Gasch et al., 2000) and proteome (Breker et al., 2013; Chong et al., 2015).”

Related to the previous issue, a quick look at the proteins tested (if I understood them correctly) showed that they were enriched for genes encoding the elongator holoenzyme complex, DNA-directed RNA polymerase I complex, membrane docking and actin binding proteins, among other functional enrichments. Genes related to DNA damage (endonuclease activity and transposition), were depleted. It was unclear whether the functional enrichment analyses described in the paper reported enrichments relative to what would be expected given the bias inherent to the tested space?

We did two functional enrichment analyses in this study: network density within Gene Ontology terms (related to Figure 2) and gene ontology enrichment of network communities (related to Figure 3). For both analyses, we performed comparisons to proteins included in PPiSeq library. This is described in the Materials and methods:

“To estimate GO term enrichment in our PPI network, we constructed 1000 random networks by replacing each bait or prey protein that was involved in a PPI with a randomly chosen protein from all proteins in our screen. This randomization preserves the degree distribution of the network.”

And:

“The set of proteins used for enrichment comparison are proteins that are involved in at least one PPI as determined by PPiSeq.”

Re: data quality. To the study's great credit, they incorporated positive and random reference sets (PRS and RRS) into the screen. However, the results from this were concerning: Appendix 1—table 1 shows that assay stringency was set such that between 1 and 3 out of 67 RRS pairs were detected. This specificity would be fine for an assay intended for retest or validate previous hits, where the prior probability of a true interaction is high, but in large-scale screening the prior probability of true interactions that are detectable by PCA is much lower, and a higher specificity is needed to avoid being overwhelmed by false positives. Consider this back of the envelope calculation: Let's say that the prior probability of true interaction is 1% as the authors' suggest, and if PCA can optimistically detect 30% of these pairs, then the number of true interactions we might expect to see in an RRS of size 67 is 1% * 30% * 67 = 0.2. This back of the envelope calculation suggests that a stringency allowing 1 hit in RRS will yield 80% [ (1 – 0.2) / 1 ] false positives, and a stringency allowing 3 hits in RRS will yield 93% [ (3 – 0.2) / 3] false positives. How do the authors reconcile these back of the envelope calculations from their PRS and RRS results with their estimates of precision?

We thank the reviewer for bringing up with this issue. We included positive and random reference sets (PRS:70 protein pairs, RRS:67 protein pairs) to benchmark our PPI calling (Yu et al., 2008). The PRS reference lists PPIs that have been validated by multiple independent studies and is therefore likely to represent true PPIs that are present in some subset of the environments we tested. For the PRS set, we found a rate of detection that is comparable to other studies (PPiSeq in SD: 28%, Y2H and yellow fluorescent protein-PCA: ~20%) (Yu et al., 2008). The RRS reference, developed ten years ago, is randomly chosen protein pairs for which there was no evidence of a PPI in the literature at the time (mostly in standard growth conditions). Given the relatively high rate of false negatives in PPI assays, this set may in fact contain some true PPIs that have yet to be discovered. We could detect PPIs for four RRS protein pairs in our study, when looking across all 9 environments. Three of these (Grs1_Pet10, Rck2_Csh1, and YDR492W_Rpd3) could be detected in multiple environments (9, 7, and 3, respectively), suggesting that their detection was not a statistical or experimental artifact of our bar-seq assay (see Author response table 1 derived from Table S4). The remaining PPI detected in the RRS, was only detected in SD (standard growth conditions) but with a relatively high fitness (0.35), again suggesting its detection was not a statistical or experimental artifact. While we do acknowledge it is possible that these are indeed false positives due to erroneous interactions of chimeric DHFR-tagged versions of these proteins, the small size of the RRS combined with the fact that some of the protein pairs could be true PPIs, did not give us confidence that this rate (4 of 70) is representative of our true false positive rate. To determine a false positive rate that is less subject to biases stemming from sampling of small numbers, we instead generated 50 new, larger random reference sets, by sampling for each set ~60,000 protein pairs without a reported PPI in BioGRID. Using these new reference sets, we found that the putative false positive rate of our assay is generally lower than 0.3% across conditions for each of the 50 reference sets. We therefore used this more statistically robust measure of the false positive rate to estimate positive predictive values (PPV = 62%, TPR = 41% in SD). We detail these statistical methods in the Appendix and report all statistical metrics in Appendix 1—table 1.

**Author response table 1. resptable1:** Mean fitness in each environment.

PPI	Environ ment_n umber	SD	H_2_O_2_	Hydroxyurea	Doxorubicin	Forskolin	Raffinose	NaCl	16℃	FK506
Rck2_Csh1	7	0.35	0.35	0	0.20	0.54	0.74	0	0.17	0.59
Grs1_Pet10	9	0.44	0.39	0.34	0.25	0.65	1.19	0.2	0.16	0.95
YDR492W_R pd3	3	0	0.18	0	0	0	0	0	0.17	0.61
Mrps35_Bub 3	1	0.35	0	0	0	0	0	0	0	0
Positive_cont rol	9	1	0.8	0.73	0.62	1.4	2.44	0.4	0.28	1.8

Methods for estimating precision and recall were not sufficiently well described to assess. Precision vs recall plots would be helpful to better understand this tradeoff as score thresholds were evaluated.

We describe in detail our approach to calling PPIs in the Appendix, including Appendix 1—table 1, and Appendix 1—figures 1, 2, 4, and now Figure 3—figure supplement 3. We identified positive PPIs using a dynamic threshold that considers the mean fitness and p-value in each environment. For each dynamic threshold, we estimated the precision and recall based on the reference sets. We then chose the threshold with the maximal Matthews correlation coefficient (MCC) to obtain the best balance between precision and recall. We have now added an additional plot (Appendix 1—figure 4) that shows the precision and recall for the chosen dynamic threshold in each environment.

Within the tested space, the Tarassov et al. map and the current map could each be compared against a common “bronze standard” (e.g. literature curated interactions), at least for the SD map, to have an idea about how the quality of the current map compares to that of the previous PCA map. Each could also be compared with the most recent large-scale Y2H study (Yu et al.).

We thank the reviewer for this suggestion. We have now added a figure panel (Figure 1—figure supplement 4) that compares PPiSeq in SD (2 replicates) to mDHFR PCA (Tarassov et al., 2008), Y2H (Yu et al., 2008), and our newly constructed “bronze standard” high-confidence positive reference set (Appendix 1 subsection “Construction of positive reference sets”).

Experimental validation of the network was done by conventional PCA. However, it should be noted that this is a form of technical replication of the DHFR-based PCA assay, and not a truly independent validation. Other large-scale yeast interaction studies (e.g., Yu et al., Science 2008) have assessed a random subset of observed PPIs using an orthogonal approach, calibrated using PRS and RRS sets examined via the same orthogonal method, from which overall performance of the dataset could be determined.

We appreciate the reviewer’s perspective, since orthogonal validation experiments have been a critical tool to establish assay performance following early Y2H work. We know from careful work done previously that modern orthogonal assays have a low cross validation rate (< 30%) (Yu et al., 2008) and that they tend to be enriched for PPIs in different cellular compartments (Jensen and Bork, 2008), indicating that high false negative rates are the likely explanation. High false negative rates have been confirmed here and elsewhere using positive reference sets (e.g. Y2H 80%, PCA 80%, PPiSeq 74% using the PRS in (Yu et al., 2008)). Therefore, the expectation is that PPiSeq, as with other assays, will have a low rate of validation using an orthogonal assay – although we would not know if this rate is 10%, 30% or somewhere in between without performing the work. However, the exact number – whether it be 10% or 30% – has no practical impact on the main conclusions of this study (focused on network dynamics rather than network enumeration). Neither does that number speak to the confidence in our PPI calls, since a lower number may simply be due to less overlap in the sets of PPIs that are callable by PPiSeq and another assay. Our method uses bar-seq to extend an established mDHFR-PCA assay (Tarassov et al., 2008). The validations we performed were aimed at confirming that our sequencing, barcode counting, fitness estimation, and PPI calling protocols were not introducing excessive noise relative to mDHFR-PCA that resulted in a high number of PPI miscalls. Confirming this, we do indeed find a high rate of validation by lower throughput PCA (50-90%, Figure 3B). Finally, we do include independent tests of the quality of our data by comparing it to positive and random reference sets from literature curated data. We find that our assay performs extremely well (PPV > 61%, TPR > 41%) relative to other high-throughput assays.

The Venn diagram in Figure 1G was not very informative in terms of assessing the quality of data. It looks like there is a relatively little overlap between PPIs identified in standard conditions (SD media) in the current study and those of the previous study using a very similar method. Is there any way to know how much of this disagreement can be attributed to each screen being sub-saturation (e.g. by comparing replica screens) and what fraction to systematic assay or environment differences?

We have now added a figure panel (Figure 3—figure supplement 1) that compares PPiSeq in SD (2 replicates) to mDHFR-PCA (Tarassov et al., 2008), Y2H (Yu et al., 2008), and our newly constructed “bronze standard” high-confidence positive reference sets (Appendix 1 subsection “Construction of positive reference sets”). We find that SD replicates have an overlap coefficient of 79% with each other, ~45% with mDHFR-PCA, ~45% the “bronze standard” PRS, and ~13% with Y2H. Overlap coefficients between the SD replicates and mDHFR-PCA are much higher than those found between orthologous methods (< 30%) or even Y2H replicates (< 35%) (Yu et al., 2008), indicating that these two assays are identifying a similar set of PPIs. We do note that PPiSeq and mDHFR-PCA do screen for PPIs under different growth conditions (batch liquid growth vs. colonies on agar), so some fraction of the disagreement is due to environmental differences. PPIs that overlap between the two PPiSeq SD replicates are more likely to be found in mDHFR-PCA, PRS, and Y2H, indicating that PPIs identified in a single SD replicate are more likely to be false positives. However, we do find (a lower rate of) overlaps between PPIs identified in only one SD replicate and other methods, suggesting that a single PPiSeq replicate is not finding all discoverable PPIs.

In Figure 3—figure supplement 2C, the environment-specificity rate of PPIs might be inflated due to the fact that authors only test for the absence of SD hits in other conditions, and the SD condition is the only condition that has been sampled twice during the screening. What would be the environment-specific verification rate if sample hits from each environment were tested in all environments? This seems important, as robustly detecting environment-specific PPIs is one of the key points of the study.

We use PPIs found in the SD environment to determine the environment-specificity because this provides the most conservative (highest) estimate of the number of PPIs found in other environments that were not detectable by our bar-seq assay. To identify PPIs in the SD environment, we pooled fitness estimates across the two replicates (~4 fitness estimates per replicate, ~ 8 total). The higher number of replicates results in a reduced rate of false positives (an erroneous fitness estimate has less impact on a PPI call), meaning that we are more confident that PPIs identified in SD are true positives. Because false positives in one environment (but not other environments) are likely to erroneously contribute to the environment-specificity rate, choosing the environment with the lowest rate of false positives (SD) should result in the lowest environment-specificity rate (highest estimate of PPIs found in other environments that were not detectable by our bar-seq assay).

Reviewer #3 (Significance):Knowledge of protein-protein interactions (PPIs) provides a key window on biological mechanism, and unbiased screens have informed global principles underlying cellular organization. Several genome-scale screens for direct (binary) interactions between yeast proteins have been carried out, and while each has provided a wealth of new hypotheses, each has been sub-saturation. Therefore, even given multiple genome-scale screens our knowledge of yeast interactions remains incomplete. Different assays are better suited to find different interactions, and it is now clear that every assay evaluated thus far is only capable (even in a saturated screen) of detecting a minority of true interactions. More relevant to the current study, no binary interaction screen has been carried out at the scale of millions of protein pairs outside of a single “baseline” condition.The study by Liu et al. is notable from a technology perspective in that it is one of several recombinant-barcode approaches have been developed to multiplex pairwise combinations of two barcoded libraries. Although other methods have been demonstrated at the scale of 1M protein pairs, this is the first study using such a technology at the scale of >1M pairs across multiple environments.A limitation is that this study is not genome-scale, and the search space is biased towards proteins for which interactions were previously observed in a particular environment. This is perhaps understandable, as it made the study more tractable, but this does add caveats to many of the conclusions drawn. These would be acceptable if clearly described and discussed. There were also questions about data quality and assessment that would need to be addressed.Assuming issues can be addressed, this is a timely study on an important topic, and will be of broad interest given the importance of protein interactions and the status of *S. cerevisiae* as a key testbed for systems biology.

References

Brauer, M.J., Huttenhower, C., Airoldi, E.M., Rosenstein, R., Matese, J.C., Gresham, D., Boer, V.M., Troyanskaya, O.G., and Botstein, D. (2007). Coordination of Growth Rate, Cell Cycle, Stress Response, and Metabolic Activity in Yeast. Mol. Biol. Cell 19, 352–367.

Breker, M., Gymrek, M., and Schuldiner, M. (2013). A novel single-cell screening platform reveals proteome plasticity during yeast stress responses. J. Cell Biol. 200, 839–850.

Chong, Y.T., Koh, J.L.Y., Friesen, H., Kaluarachchi Duffy, S., Cox, M.J., Moses, A., Moffat, J., Boone, C., and Andrews, B.J. (2015). Yeast Proteome Dynamics from Single Cell Imaging and Automated Analysis. Cell 161, 1413–1424.

Gasch, A.P., Spellman, P.T., Kao, C.M., Carmel-Harel, O., Eisen, M.B., Storz, G., Botstein, D., and Brown, P.O. (2000). Genomic Expression Programs in the Response of Yeast Cells to Environmental Changes. Mol. Biol. Cell 11, 4241–4257.

Hart, G.T., Ramani, A.K., and Marcotte, E.M. (2006). How complete are current yeast and human protein-interaction networks? Genome Biol. 7, 120.

Hilliker, A., Gao, Z., Jankowsky, E., and Parker, R. (2011). The DEAD-box protein Ded1 modulates translation by the formation and resolution of an eIF4F-mRNA complex. Mol. Cell 43, 962–972.

Isasa, M., Suñer, C., Díaz, M., Puig-Sàrries, P., Zuin, A., Bichmann, A., Gygi, S.P., Rebollo, E., and Crosas, B. (2015). Cold Temperature Induces the Reprogramming of Proteolytic Pathways in Yeast. J. Biol. Chem. jbc.M115.698662.

Jensen, L.J., and Bork, P. (2008). Not Comparable, But Complementary. Science 322, 56–57.

Lahtvee, P.-J., Sánchez, B.J., Smialowska, A., Kasvandik, S., Elsemman, I.E., Gatto, F., and Nielsen, J. (2017). Absolute Quantification of Protein and mRNA Abundances Demonstrate Variability in Gene-Specific Translation Efficiency in Yeast. Cell Syst. 4, 495-504.e5.

Obayashi, T., Kagaya, Y., Aoki, Y., Tadaka, S., and Kinoshita, K. (2019). COXPRESdb v7: a gene coexpression database for 11 animal species supported by 23 coexpression platforms for technical evaluation and evolutionary inference. Nucleic Acids Res. 47, D55–D62.

Sambourg, L., and Thierry-Mieg, N. (2010). New insights into protein-protein interaction data lead to increased estimates of the *S. cerevisiae* interactome size. BMC Bioinformatics 11, 605.

Tarassov, K., Messier, V., Landry, C.R., Radinovic, S., Molina, M.M.S., Shames, I., Malitskaya, Y., Vogel, J., Bussey, H., and Michnick, S.W. (2008). An in Vivo Map of the Yeast Protein Interactome. Science 320, 1465–1470.

Yu, H., Braun, P., Yıldırım, M.A., Lemmens, I., Venkatesan, K., Sahalie, J., Hirozane-Kishikawa, T., Gebreab, F., Li, N., Simonis, N., et al. (2008). High-Quality Binary Protein Interaction Map of the Yeast Interactome Network. Science 322 , 104–110.